# Finite-Time Analysis of Whittle Index based Q-Learning for Restless Multi-Armed Bandits with Neural Network Function Approximation

**Guojun Xiong,  Jian Li**
Stony Brook University
{guojun.xiong,jian.li.3}@stonybrook.edu

## Abstract

Whittle index policy is a heuristic to the intractable restless multi-armed bandits (RMAB) problem. Although it is provably asymptotically optimal, finding Whittle indices remains difficult. In this paper, we present `Neural-Q-Whittle`, a Whittle index based Q-learning algorithm for RMAB with neural network function approximation, which is an example of nonlinear two-timescale stochastic approximation with Q-function values updated on a faster timescale and Whittle indices on a slower timescale. Despite the empirical success of deep Q-learning, the non-asymptotic convergence rate of `Neural-Q-Whittle`, which couples neural networks with two-timescale Q-learning largely remains unclear. This paper provides a finite-time analysis of `Neural-Q-Whittle`, where data are generated from a Markov chain, and Q-function is approximated by a ReLU neural network. Our analysis leverages a Lyapunov drift approach to capture the evolution of two coupled parameters, and the nonlinearity in value function approximation further requires us to characterize the approximation error. Combing these provide `Neural-Q-Whittle` with $\mathcal{O}(1/k^{2/3})$ convergence rate, where $k$ is the number of iterations.

## 1 Introduction

We consider the restless multi-armed bandits (RMAB) problem [56], where the decision maker (DM) repeatedly activates $K$ out of $N$ arms at each decision epoch. Each arm is described by a Markov decision process (MDP) [45], and evolves stochastically according to two different transition kernels, depending on whether the arm is activated or not. Rewards are generated with each transition. Although RMAB has been widely used to study constrained sequential decision making problems [5, 38, 12, 64, 29, 36, 35, 25], it is notoriously intractable due to the explosion of state space [44]. A celebrated heuristic is the Whittle index policy [56], which computes the Whittle index for each arm given its current state as the cost to pull the arm. Whittle index policy then activates the $K$ highest indexed arms at each decision epoch, and is provably asymptotically optimal [54].

However, the computation of Whittle index requires full knowledge of the underlying MDP associated with each arm, which is often unavailable in practice. To this end, many recent efforts have focused on learning Whittle indices for making decisions in an online manner. First, model-free reinforcement learning (RL) solutions have been proposed [10, 23, 53, 8, 28, 57, 59, 61, 58, 3], among which [3] developed a Whittle index based Q-learning algorithm, which we call `Q-Whittle` for ease of exposition, and provided the first-ever rigorous asymptotic analysis. However, `Q-Whittle` suffers from slow convergence since it only updates the Whittle index of a specific state when that state is visited. In addition, `Q-Whittle` needs to store the Q-function values for all state-action pairs, which limits its applicability only to problems with small state space. Second, deep RL methods have been leveraged to predict Whittle indices via training neural networks [40, 41]. Though

37th Conference on Neural Information Processing Systems (NeurIPS 2023).

these methods are capable of dealing with large state space, there is no asymptotic or finite-time performance guarantee. Furthermore, training neural networks requires to tuning hyper-parameters. This introduces an additional layer of complexity to predict Whittle indices. Third, to address aforementioned deficiencies, [60] proposed `Q-Whittle-LFA` by coupling `Q-Whittle` with linear function approximation and provided a finite-time convergence analysis. One key limitation of `Q-Whittle-LFA` is the unrealistic assumption that all data used in `Q-Whittle-LFA` are sampled i.i.d. from a fixed stationary distribution.

To tackle the aforementioned limitations and inspired by the empirical success of deep Q-learning in numerous applications, we develop `Neural-Q-Whittle`, a Whittle index based Q-learning algorithm with *neural network function approximation under Markovian observations*. Like [3, 60], the updates of Q-function values and Whittle indices form a two-timescale stochastic approximation (2TSA) with the former operating on a faster timescale and the later on a slower timescale. Unlike [3, 60], our `Neural-Q-Whittle` uses a deep neural network with the ReLU activation function to approximate the Q-function. However, Q-learning with neural network function approximation can in general diverge [2], and the theoretical convergence of Q-learning with neural network function approximation has been limited to special cases such as fitted Q-iteration with i.i.d. observations [22], which fails to capture the practical setting of Q-learning with neural network function approximation.

In this paper, we study the non-asymptotic convergence of `Neural-Q-Whittle` with data generated from a Markov decision process. Compared with recent theoretical works for Q-learning with neural network function approximation [13, 22, 62], our `Neural-Q-Whittle` involves a two-timescale update between two coupled parameters, i.e., Q-function values and Whittle indices. This renders existing finite-time analysis in [13, 22, 62] not applicable to our `Neural-Q-Whittle` due to the fact that [13, 22, 62] only contains a single-timescale update on Q-function values. Furthermore, [13, 22, 62] required an additional projection step for the update of parameters of neural network function so as to guarantee the boundedness between the unknown parameter at any time step with the initialization. This in some cases is impractical. Hence, a natural question that arises is

> *Is it possible to provide a non-asymptotic convergence rate analysis of `Neural-Q-Whittle` with two coupled parameters updated in two timescales under Markovian observations without the extra projection step?*

The theoretical convergence guarantee of two-timescale Q-learning with neural network function approximation under Markovian observations remains largely an open problem, and in this paper, we provide an affirmative answer to this question. Our main contributions are summarized as follows:

• We propose `Neural-Q-Whittle`, a novel Whittle index based Q-learning algorithm with neural network function approximation for RMAB. Inspired by recent work on TD learning [48] and Q-learning [15] with linear function approximation, our `Neural-Q-Whittle` removes the additional impractical projection step in the neural network function parameter update.

• We establish the first finite-time analysis of `Neural-Q-Whittle` under Markovian observations. Due to the two-timescale nature for the updates of two coupled parameters (i.e., Q-function values and Whittle indices) in `Neural-Q-Whittle`, we focus on the convergence rate of these parameters rather than the convergence rate of approximated Q-functions as in [13, 22, 62]. Our key technique is to view `Neural-Q-Whittle` as a 2TSA for finding the solution of suitable nonlinear equations. Different from recent works on finite-time analysis of a general 2TSA [20] or with linear function approximation [60], the nonlinear parameterization of Q-function in `Neural-Q-Whittle` under Markovian observations imposes significant difficulty in finding the global optimum of the corresponding nonlinear equations. To mitigate this, we first approximate the original neural network function with a collection of local linearization and focus on finding a surrogate Q-function in the neural network function class that well approximates the optimum. Our finite-time analysis then requires us to consider two Lyapunov functions that carefully characterize the coupling between iterates of Q-function values and Whittle indices, with one Lyapunov function defined with respect to the true neural network function, and the other defined with respect to the locally linearized neural network function. We then characterize the errors between these two Lyapunov functions. Putting them together, we prove that `Neural-Q-Whittle` achieves a convergence in expectation at a rate $\mathcal{O}(1/k^{2/3})$, where $k$ is the number of iterations.

• Finally, we conduct experiments to validate the convergence performance of `Neural-Q-Whittle`, and verify the sufficiency of our proposed condition for the stability of `Neural-Q-Whittle`.

## 2 Preliminaries

**RMAB.** We consider an infinite-horizon average-reward RMAB with each arm $n \in \mathcal{N}$ described by a unichain MDP [45] $\mathcal{M}_n := (\mathcal{S}, \mathcal{A}, P_n, r_n)$, where $\mathcal{S}$ is the state space with cardinality $S < \infty$, $\mathcal{A}$ is the action space with cardinality $A$, $P_n(s'|s, a)$ is the transition probability of reaching state $s'$ by taking action $a$ in state $s$, and $r_n(s, a)$ is the reward associated with state-action pair $(s, a)$. At each time slot $t$, the DM activates $K$ out of $N$ arms. Arm $n$ is "active" at time $t$ when it is activated, i.e., $A_n(t) = 1$; otherwise, arm $n$ is "passive", i.e., $A_n(t) = 0$. Let $\Pi$ be the set of all possible policies for RMAB, and $\pi \in \Pi$ is a feasible policy, satisfying $\pi : \mathcal{F}_t \mapsto \mathcal{A}^N$, where $\mathcal{F}_t$ is the sigma-algebra generated by random variables $\{S_n(h), A_n(h) : \forall n \in \mathcal{N}, h \le t\}$. The objective of the DM is to maximize the expected long-term average reward subject to an instantaneous constraint that only $K$ arms can be activated at each time slot, i.e.,

$$\text{RMAB:} \quad \max_{\pi \in \Pi} \liminf_{T \to \infty} \frac{1}{T} \mathbb{E}_\pi \left( \sum_{t=0}^{T} \sum_{n=1}^{N} r_n(t) \right), \quad \text{s.t.} \sum_{n=1}^{N} A_n(t) = K, \quad \forall t. \quad (1)$$

**Whittle Index Policy.** It is well known that RMAB (1) suffers from the curse of dimensionality [44]. To address this challenge, Whittle [56] proposed an index policy through decomposition. Specifically, Whittle relaxed the constraint in (1) to be satisfied on average and obtained a unconstrained problem: $\max_{\pi \in \Pi} \liminf_{T \to \infty} \frac{1}{T} \mathbb{E}_\pi \sum_{t=1}^{T} \sum_{n=1}^{N} \{r_n(t) + \lambda(1 - A_n(t))\}$, where $\lambda$ is the Lagrangian multiplier associated with the constraint. The key observation of Whittle is that this problem can be decomposed and its solution is obtained by combining solutions of $N$ independent problems via solving the associated dynamic programming (DP): $V_n(s) = \max_{a \in \{0,1\}} Q_n(s, a), \forall n \in \mathcal{N}$, where

$$Q_n(s,a) + \beta = a\left(r_n(s,a) + \sum_{s'} p_n(s'|s,1)V_n(s')\right) + (1-a)\left(r_n(s,a) + \lambda + \sum_{s'} p_n(s'|s,0)V_n(s')\right), \quad (2)$$

where $\beta$ is unique and equals to the maximal long-term average reward of the unichain MDP, and $V_n(s)$ is unique up to an additive constant, both of which depend on the Lagrangian multiplier $\lambda$. The optimal decision $a^*$ in state $s$ then is the one which maximizes the right hand side of the above DP. The Whittle index associated with state $s$ is defined as the value $\lambda_n^*(s) \in \mathbb{R}$ such that actions 0 and 1 are equally favorable in state $s$ for arm $n$ [3, 23], satisfying

$$\lambda_n^*(s) := r_n(s, 1) + \sum_{s'} p_n(s'|s, 1)V_n(s') - r_n(s, 0) - \sum_{s'} p_n(s'|s, 0)V_n(s'). \quad (3)$$

Whittle index policy then activates $K$ arms with the largest Whittle indices at each time slot. Additional discussions are provided in Section B in supplementary materials.

**Q-Learning for Whittle Index.** Since the underlying MDPs are often unknown, [3] proposed `Q-Whittle`, a tabular Whittle index based Q-learning algorithm, where the updates of Q-function values and Whittle indices form a 2TSA, with the former operating on a faster timescale for a given $\lambda_n$ and the later on a slower timescale. Specifically, the Q-function values for $\forall n \in \mathcal{N}$ are updated as

$$Q_{n,k+1}(s,a) := Q_{n,k}(s,a) + \alpha_{n,k} \mathbb{1}_{\{S_{n,k}=s, A_{n,k}=a\}} \Big( r_n(s,a) + (1-a)\lambda_{n,k}(s)$$
$$+ \max_a Q_{n,k}(S_{n,k+1}, a) - I_n(Q_k) - Q_{n,k}(s,a) \Big), \quad (4)$$

where $I_n(Q_k) = \frac{1}{2S} \sum_{s \in \mathcal{S}} (Q_{n,k}(s, 0) + Q_{n,k}(s, 1))$ is standard in the relative Q-learning for long-term average MDP setting [1], which differs significantly from the discounted reward setting [45, 1]. $\{\alpha_{n,k}\}$ is a step-size sequence satisfying $\sum_k \alpha_{n,k} = \infty$ and $\sum_k \alpha_{n,k}^2 < \infty$.

Accordingly, the Whittle index is updated as

$$\lambda_{n,k+1}(s) = \lambda_{n,k}(s) + \eta_{n,k}(Q_{n,k}(s, 1) - Q_{n,k}(s, 0)), \quad (5)$$

with the step-size sequence $\{\eta_{n,k}\}$ satisfying $\sum_k \eta_{n,k} = \infty$, $\sum_k \eta_{n,k}^2 < \infty$ and $\eta_{n,k} = o(\alpha_{n,k})$. The coupled iterates (4) and (5) form a 2TSA, and [3] provided an asymptotic convergence analysis.

## 3 Neural Q-Learning for Whittle Index

A closer look at (5) reveals that `Q-Whittle` only updates the Whittle index of a specific state when that state is visited. This makes `Q-Whittle` suffers from slow convergence. In addition, `Q-Whittle`

---

**Algorithm 1** `Neural-Q-Whittle`: Neural Q-Learning for Whittle Index

---

1: **Input**: $\boldsymbol{\phi}(s,a)$ for $\forall s \in \mathcal{S}, a \in \mathcal{A}$, and learning rates $\{\alpha_k\}_{k=1,\dots,T}, \{\eta_k\}_{k=1,\dots,T}$
2: **Initialization**: $b_r \sim \text{Unif}(\{-1,1\}), \mathbf{w}_{r,0} \sim \mathcal{N}(\mathbf{0}, \mathbf{I}_d/d), \forall r \in [1,m]$ and $\lambda(s) = 0, \forall s \in \mathcal{S}$
3: **for** $s \in \mathcal{S}$ **do**
4:    **for** $k = 1,\dots,T$ **do**
5:       Sample $(S_k, A_k, S_{k+1})$ according to the $\epsilon$-greedy policy;
6:       $\Delta_k = r(S_k, A_k) + (1-A_k)\lambda_k(s) + \max_a f(\boldsymbol{\theta}_k; \boldsymbol{\phi}(S_{k+1}, a)) - I(\boldsymbol{\theta}_k) - f(\boldsymbol{\theta}_k; \boldsymbol{\phi}(S_k, A_k));$
7:       $\boldsymbol{\theta}_{k+1} = \boldsymbol{\theta}_k + \alpha_k \Delta_k \nabla_{\boldsymbol{\theta}} f(\boldsymbol{\theta}_k; \boldsymbol{\phi}(S_k, A_k));$
8:       $\lambda_{k+1}(s) = \lambda_k(s) + \eta_k(f(\boldsymbol{\theta}_k; \boldsymbol{\phi}(s,1)) - f(\boldsymbol{\theta}_k; \boldsymbol{\phi}(s,0));$
9:    **end for**
10: **end for**
11: **Return:** $\lambda(s), \forall s \in \mathcal{S}$.

---

needs to store the Q-function values for all state-action pairs, which limits its applicability only to problems with small state space. To address this challenge and inspired by the empirical success of deep Q-learning, we develop `Neural-Q-Whittle` through coupling `Q-Whittle` with neural network function approximation by using low-dimensional feature mapping and leveraging the strong representation power of neural networks. For ease of presentation, we drop the subscript $n$ in (4) and (5), and discussions in the rest of the paper apply to any arm $n \in \mathcal{N}$.

Specifically, given a set of basis functions $\phi_\ell : \mathcal{S} \times \mathcal{A} \mapsto \mathbb{R}, \forall \ell = 1, \cdots, d$ with $d \ll SA$, the approximation of Q-function $Q_{\boldsymbol{\theta}}(s,a)$ parameterized by a unknown weight vector $\boldsymbol{\theta} \in \mathbb{R}^{md}$, is given by $Q_{\boldsymbol{\theta}}(s,a) = f(\boldsymbol{\theta}; \boldsymbol{\phi}(s,a)), \forall s \in \mathcal{S}, a \in \mathcal{A}$, where $f$ is a nonlinear neural network function parameterized by $\boldsymbol{\theta}$ and $\boldsymbol{\phi}(s,a)$, with $\boldsymbol{\phi}(s,a) = (\phi_1(s,a), \cdots, \phi_d(s,a))^{\mathsf{T}}$. The feature vectors are assumed to be linearly independent and are normalized so that $\|\boldsymbol{\phi}(s,a)\| \le 1, \forall s \in \mathcal{S}, a \in \mathcal{A}$. In particular, we parameterize the Q-function by using a two-layer neural network [13, 62]

$$f(\boldsymbol{\theta}; \boldsymbol{\phi}(s,a)) := \frac{1}{\sqrt{m}} \sum_{r=1}^m b_r \sigma(\mathbf{w}_r^{\mathsf{T}} \boldsymbol{\phi}(s,a)), \tag{6}$$

where $\boldsymbol{\theta} = (b_1, \dots, b_m, \mathbf{w}_1^{\mathsf{T}}, \dots, \mathbf{w}_m^{\mathsf{T}})^{\mathsf{T}}$ with $b_r \in \mathbb{R}$ and $\mathbf{w}_r \in \mathbb{R}^{d \times 1}, \forall r \in [1, m]$. $b_r, \forall r$ are uniformly initialized in $\{-1, 1\}$ and $w_r, \forall r$ are initialized as a zero mean Gaussian distribution according to $\mathcal{N}(\mathbf{0}, \mathbf{I}_d/d)$. During training process, only $\mathbf{w}_r, \forall r$ are updated while $b_r, \forall r$ are fixed as the random initialization. Hence, we use $\boldsymbol{\theta}$ and $\mathbf{w}_r, \forall r$ interchangeably throughout this paper. $\sigma(x) = \max(0, x)$ is the rectified linear unit (ReLU) activation function[1].

Given (6), we can rewrite the Q-function value updates in (4) as

$$\boldsymbol{\theta}_{k+1} = \boldsymbol{\theta}_k + \alpha_k \Delta_k \nabla_{\boldsymbol{\theta}} f(\boldsymbol{\theta}_k; \boldsymbol{\phi}(S_k, A_k)), \tag{7}$$

with $\Delta_k$ being the temporal difference (TD) error defined as $\Delta_k := r(S_k, A_k) + (1 - A_k)\lambda_k(s) - I(\boldsymbol{\theta}_k) + \max_a f(\boldsymbol{\theta}_k; \boldsymbol{\phi}(S_{k+1}, a)) - f(\boldsymbol{\theta}_k; \boldsymbol{\phi}(S_k, A_k))$, where $I(\boldsymbol{\theta}_k) = \frac{1}{2S} \sum_{s \in \mathcal{S}} [f(\boldsymbol{\theta}_k; \boldsymbol{\phi}(s,0)) + f(\boldsymbol{\theta}_k; \boldsymbol{\phi}(s,1))]$. Similarly, the Whittle index update (5) can be rewritten as

$$\lambda_{k+1}(s) = \lambda_k(s) + \eta_k(f(\boldsymbol{\theta}_k; \boldsymbol{\phi}(s,1)) - f(\boldsymbol{\theta}_k; \boldsymbol{\phi}(s,0))). \tag{8}$$

The coupled iterates in (7) and (8) form `Neural-Q-Whittle` as summarized in Algorithm 1, which aims to learn the coupled parameters $(\boldsymbol{\theta}^*, \lambda^*(s))$ such that $f(\boldsymbol{\theta}^*, \boldsymbol{\phi}(s,1)) = f(\boldsymbol{\theta}^*, \boldsymbol{\phi}(s,0)), \forall s \in \mathcal{S}$.

**Remark 1.** *Unlike recent works for Q-learning with linear [6, 37, 67] or neural network function approximations [13, 22, 62], we do not assume an additional projection step of the updates of unknown parameters $\boldsymbol{\theta}_k$ in (7) to confine $\boldsymbol{\theta}_k, \forall k$ into a bounded set. This projection step is often used to stabilize the iterates related to the unknown stationary distribution of the underlying Markov chain, which in some cases is impractical. More recently, [48] removed the extra projection step and established the finite-time convergence of TD learning, which is treated as a linear stochastic approximation algorithm. [15] extended it to the Q-learning with linear function approximation.*

---

[1]The finite-time analysis of Deep Q-Networks (DQN) [13, 22, 62] and references therein focuses on the ReLU activation function, as it has certain properties that make the analysis tractable. ReLU is piecewise linear and non-saturating, which can simplify the mathematical analysis. Applying the same analysis to other activation functions like the hyperbolic tangent (tanh) could be more complex, which is out of the scope of this work.

Table 1: Comparison of settings in related works.

| Algorithm | Noise | Approximation | Timescale | Whittle index |
|---|---|---|---|---|
| Q-Whittle [3] | *i.i.d.* | ✗ | *two-timescale* | ✓ |
| Q-Whittle-LFA [60] | *i.i.d.* | linear | *two-timescale* | ✓ |
| Q-Learning-LFA [6, 37, 67] | *Markovian* | linear | *single-timescale* | ✗ |
| Q-Learning-NFA [13, 15, 22, 62] | *Markovian* | *neural network* | *single-timescale* | ✗ |
| TD-Learning-LFA [48] | *Markovian* | linear | *single-timescale* | ✗ |
| 2TSA-IID [19, 21] | *i.i.d.* | ✗ | *two-timescale* | ✗ |
| 2TSA-Markovian [20] | *Markovian* | ✗ | *two-timescale* | ✗ |
| Neural-Q-Whittle (**this work**) | *Markovian* | *neural network* | *two-timescale* | ✓ |

*However, these state-of-the-art works only contained a single-timescale update on Q-function values, i.e., with the only unknown parameter $\boldsymbol{\theta}$, while our* Neural-Q-Whittle *involves a two-timescale update between two coupled unknown parameters $\boldsymbol{\theta}$ and $\lambda$ as in (7) and (8). Our goal in this paper is to expand the frontier by providing a finite-time bound for* Neural-Q-Whittle *under Markovian noise without requiring an additional projection step. We summarize the differences between our work and existing literature in Table 1.*

## 4 Finite-Time Analysis of Neural-Q-Whittle

In this section, we present the finite-time analysis of Neural-Q-Whittle for learning Whittle index $\lambda(s)$ of any state $s \in \mathcal{S}$ when data are generated from a MDP. To simplify notation, we abbreviate $\lambda(s)$ as $\lambda$ in the rest of the paper. We start by first rewriting the updates of Neural-Q-Whittle in (7) and (8) as a nonlinear two-timescale stochastic approximation (2TSA) in Section 4.1.

### 4.1 A Nonlinear 2TSA Formulation with Neural Network Function

We first show that Neural-Q-Whittle can be rewritten as a variant of the nonlinear 2TSA. For any fixed policy $\pi$, since the state of each arm $\{S_k\}$ evolves according to a Markov chain, we can construct a new variable $X_k = (S_k, A_k, S_{k+1})$, which also forms a Markov chain with state space $\mathcal{X} := \{(s, a, s') | s \in \mathcal{S}, \pi(a|s) \geq 0, p(s'|s, a) > 0\}$. Therefore, the coupled updates (7) and (8) of Neural-Q-Whittle can be rewritten in the form of a nonlinear 2TSA [20]:

$$\boldsymbol{\theta}_{k+1} = \boldsymbol{\theta}_k + \alpha_k h(X_k, \boldsymbol{\theta}_k, \lambda_k), \qquad \lambda_{k+1} = \lambda_k + \eta_k g(X_k, \boldsymbol{\theta}_k, \lambda_k), \tag{9}$$

where $\boldsymbol{\theta}_0$ and $\lambda_0$ being arbitrarily initialized in $\mathbb{R}^{md}$ and $\mathbb{R}$, respectively; and $h(\cdot)$ and $g(\cdot)$ satisfy

$$h(X_k, \boldsymbol{\theta}_k, \lambda_k) := \nabla_{\boldsymbol{\theta}} f(\boldsymbol{\theta}_k; \boldsymbol{\phi}(S_k, A_k)) \Delta_k, \qquad \boldsymbol{\theta}_k \in \mathbb{R}^{md}, \lambda_k \in \mathbb{R}, \tag{10}$$

$$g(X_k, \boldsymbol{\theta}_k, \lambda_k) := f(\boldsymbol{\theta}_k; \boldsymbol{\phi}(s, 1)) - f(\boldsymbol{\theta}_k; \boldsymbol{\phi}(s, 0)), \qquad \boldsymbol{\theta}_k \in \mathbb{R}^{md}. \tag{11}$$

Since $\eta_k \ll \alpha_k$, the dynamics of $\boldsymbol{\theta}$ evolves much faster than those of $\lambda$. We aim to establish the finite-time performance of the nonlinear 2TSA in (9), where $f(\cdot)$ is the neural network function defined in (6). This is equivalent to find the root[2] $(\boldsymbol{\theta}^*, \lambda^*)$ of a system with *two coupled* nonlinear equations $h : \mathcal{X} \times \mathbb{R}^{md} \times \mathbb{R} \rightarrow \mathbb{R}^{md}$ and $g : \mathcal{X} \times \mathbb{R}^{md} \times \mathbb{R} \rightarrow \mathbb{R}$ such that

$$H(\boldsymbol{\theta}, \lambda) := \mathbb{E}_{\mu}[h(X, \boldsymbol{\theta}, \lambda)] = 0, \qquad G(\boldsymbol{\theta}, \lambda) := \mathbb{E}_{\mu}[g(X, \boldsymbol{\theta}, \lambda)] = 0, \tag{12}$$

where $X$ is a random variable in finite state space $\mathcal{X}$ with unknown distribution $\mu$. For a fixed $\boldsymbol{\theta}$, to study the stability of $\lambda$, we assume the condition on the existence of a mapping such that $\lambda = y(\boldsymbol{\theta})$ is the unique solution of $G(\boldsymbol{\theta}, \lambda) = 0$. In particular, $y(\boldsymbol{\theta})$ is given as

$$y(\boldsymbol{\theta}) = r(s,1) + \sum_{s'} p(s'|s,1) \max_a f(\boldsymbol{\theta}; \boldsymbol{\phi}(s', a)) - r(s,0) - \sum_{s'} p(s'|s,0) \max_a f(\boldsymbol{\theta}; \boldsymbol{\phi}(s', a)). \tag{13}$$

---

[2]The root $(\boldsymbol{\theta}^*, \lambda^*)$ of the nonlinear 2TSA (9) can be established by using the ODE method following the solution of suitably defined differential equations [9, 49, 3, 21, 19, 20], i.e., $\dot{\boldsymbol{\theta}} = H(\boldsymbol{\theta}, \lambda), \dot{\lambda} = \frac{\eta}{\alpha} G(\boldsymbol{\theta}, \lambda)$, where a fixed stepsize is assumed for ease of expression at this moment.

## 4.2 Main Results

As inspired by [20], the finite-time analysis of such a nonlinear 2TSA boils down to the choice of two step sizes $\{\alpha_k, \eta_k, \forall k\}$ and a Lyapunov function that couples the two iterates in (9). To this end, we first define the following two error terms:

$$\tilde{\boldsymbol{\theta}}_k = \boldsymbol{\theta}_k - \boldsymbol{\theta}^*, \qquad \tilde{\lambda}_k = \lambda_k - y(\boldsymbol{\theta}_k), \tag{14}$$

which characterize the coupling between $\boldsymbol{\theta}_k$ and $\lambda_k$. If $\tilde{\boldsymbol{\theta}}_k$ and $\tilde{\lambda}_k$ go to zero simultaneously, the convergence of $(\boldsymbol{\theta}_k, \lambda_k)$ to $(\boldsymbol{\theta}^*, \lambda^*)$ can be established. Thus, to prove the convergence of $(\boldsymbol{\theta}_k, \lambda_k)$ of the nonlinear 2TSA in (9) to its true value $(\boldsymbol{\theta}^*, \lambda^*)$, we can equivalently study the convergence of $(\tilde{\boldsymbol{\theta}}_k, \tilde{\lambda}_k)$ by providing the finite-time analysis for the mean squared error generated by (9). To couple the fast and slow iterates, we define the following weighted Lyapunov function

$$M(\boldsymbol{\theta}_k, \lambda_k) := \frac{\eta_k}{\alpha_k} \|\tilde{\boldsymbol{\theta}}_k\|^2 + \|\tilde{\lambda}_k\|^2 = \frac{\eta_k}{\alpha_k} \|\boldsymbol{\theta}_k - \boldsymbol{\theta}^*\|^2 + \|\lambda_k - y(\boldsymbol{\theta}_k)\|^2, \tag{15}$$

where $\| \cdot \|$ stands for the the Euclidean norm for vectors throughout the paper. It is clear that the Lyapunov function $M(\boldsymbol{\theta}_k, \lambda_k)$ combines the updates of $\boldsymbol{\theta}$ and $\lambda$ with respect to the true neural network function $f(\boldsymbol{\theta}; \boldsymbol{\phi}(s, a))$ in (6).

To this end, our goal turns to characterize finite-time convergence of $\mathbb{E}[M(\boldsymbol{\theta}_k, \lambda_k)]$. However, it is challenging to directly finding the global optimum of the corresponding nonlinear equations due to the nonlinear parameterization of Q-function in `Neural-Q-Whittle`. In addition, the operators $h(\cdot), g(\cdot)$ and $y(\cdot)$ in (10), (11) and (13) directly relate with the convoluted neural network function $f(\boldsymbol{\theta}; \boldsymbol{\phi}(s, a))$ in (6), which hinders us to characterize the smoothness properties of theses operators. Such properties are often required for the analysis of stochastic approximation [15, 19, 21].

To mitigate this, **(Step 1)** we instead approximate the true neural network function $f(\boldsymbol{\theta}, \boldsymbol{\phi}(s, a))$ with a collection of local linearization $f_0(\boldsymbol{\theta}; \boldsymbol{\phi}(s, a))$ at the initial point $\boldsymbol{\theta}_0$. Based on the surrogate stationary point $\boldsymbol{\theta}_0^*$ of $f_0(\boldsymbol{\theta}; \boldsymbol{\phi}(s, a))$, we correspondingly define a modified Lyapunov function $\hat{M}(\boldsymbol{\theta}_k, \lambda_k)$ combining updates of $\boldsymbol{\theta}$ and $\lambda$ with respect to such local linearization. Specifically, we have

$$\hat{M}(\boldsymbol{\theta}_k, \lambda_k) := \frac{\eta_k}{\alpha_k} \|\boldsymbol{\theta}_k - \boldsymbol{\theta}_0^*\|^2 + \|\lambda_k - y_0(\boldsymbol{\theta}_k)\|^2, \tag{16}$$

where $y_0(\cdot)$ is in the same expression as $y(\cdot)$ in (13) by replacing $f(\cdot)$ with $f_0(\cdot)$, and we will describe this in details below. **(Step 2)** We then study the convergence rate of the nonlinear 2TSA using this modified Lyapunov function under general conditions. **(Step 3)** Finally, since the two coupled parameters $\boldsymbol{\theta}$ and $\lambda$ in (9) are updated with respect to the true neural network function $f(\boldsymbol{\theta}; \boldsymbol{\phi}(s, a))$ in (6) in `Neural-Q-Whittle`, while we characterize their convergence using the approximated neural network function in Step 2. Hence, this further requires us to characterize the approximation errors. We visualize the above three steps in Figure 1 and provide a proof sketch in Section 4.3. Combing them together gives rise to our main theoretical results on the finite-time performance of `Neural-Q-Whittle`, which is formally stated in the following theorem.

**Theorem 1.** *Consider iterates $\{\boldsymbol{\theta}_k\}$ and $\{\lambda_k\}$ generated by `Neural-Q-Whittle` in (7) and (8). Given $\alpha_k = \frac{\alpha_0}{(k+1)}, \eta_k = \frac{\eta_0}{(k+1)^{4/3}}$, we have for $\forall k \geq \tau$*

$$\mathbb{E}[M(\boldsymbol{\theta}_{k+1}, \lambda_{k+1})|\mathcal{F}_{k-\tau}] \leq \frac{2\tau^2 \mathbb{E}[\hat{M}(\boldsymbol{\theta}_\tau, \lambda_\tau)]}{(k+1)^2} + \frac{1200\alpha_0^3}{\eta_0} \frac{(C_1 + \|\hat{\boldsymbol{\theta}}_0\|)^2 + (2C_1 + \|\hat{\lambda}_0\|)^2}{(k+1)^{2/3}}$$

$$+ \frac{2\eta_0 c_0^2}{\alpha_0 (1-\kappa)^2} \|span(\Pi_{\mathcal{F}} f(\boldsymbol{\theta}^*) - f(\boldsymbol{\theta}^*))\|^2 + \left(\frac{2}{(k+1)^{2/3}} + 2\right) \mathcal{O}\left(\frac{c_1^3 (\|\boldsymbol{\theta}_0\| + |\lambda_0| + 1)^3}{m^{1/2}}\right), \tag{17}$$

*where $C_1 := c_1(\|\boldsymbol{\theta}_0\| + \|\lambda_0\| + 1)$ with $c_1$ being a proper chosen constant, $c_0$ is a constant defined in Assumption 3, $\tau$ is the mixing time defined in (22), $span$ denotes for the span semi-norm [47], and $\Pi_{\mathcal{F}}$ represents the projection to the set of $\mathcal{F}$ contianing all possible $f_0(\boldsymbol{\theta}; \boldsymbol{\phi}(s, a))$ in (18).*

The first term on the right hand side (17) corresponds to the bias compared to the Lyapunov function at the mixing time $\tau$, which goes to zero at a rate of $\mathcal{O}(1/k^2)$. The second term corresponds to the accumulated estimation error of the nonlinear 2TSA due to Markovian noise, which vanishes at the rate $\mathcal{O}(1/k^{2/3})$. Hence it dominates the overall convergence rate in (17). The third term captures the distance between the optimal solution $(\boldsymbol{\theta}^*, \lambda^*)$ to the true neural network function $f(\boldsymbol{\theta}_k; \boldsymbol{\phi}(s, a))$ in

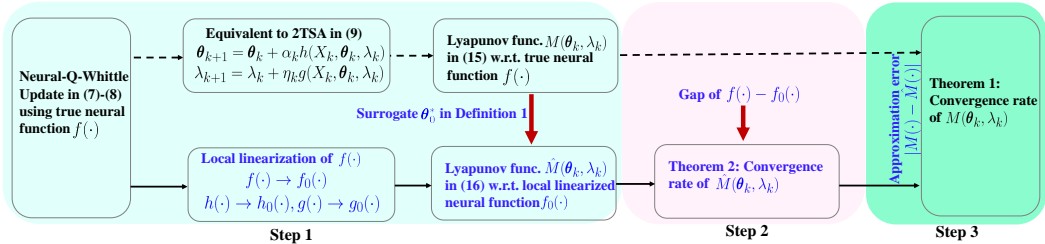

Figure 1: `Neural-Q-Whittle` operates w.r.t. true neural function $f(\cdot)$ with its finite-time performance given in Theorem 1 (indicated in dashed lines). Our proofs operate in three steps: (i) Step 1: Obtain local linearization $f_0(\cdot)$ and define Lyapunov function $\hat{M}(\cdot)$ w.r.t. $f_0(\cdot)$. (ii) Step 2: Characterize the finite-time performance w.r.t. $\hat{M}(\cdot)$ using Lyapunov drift method. Since `Neural-Q-Whittle` is updated w.r.t. $f(\cdot)$, we need to characterize the gap between $f(\cdot)$ and $f_0(\cdot)$. (iii) Step 3: Similarly, we characterize the approximation errors between $M(\cdot)$ and $\hat{M}(\cdot)$.

(6) and the optimal one $(\boldsymbol{\theta}_0^*, y_0(\boldsymbol{\theta}_0^*))$ with local linearization $f_0(\boldsymbol{\theta}_k; \boldsymbol{\phi}(s,a))$ in (18), which quantifies the error when $f(\boldsymbol{\theta}^*)$ does not fall into the function class $\mathcal{F}$. The last term characterizes the distance between $f(\boldsymbol{\theta}_k; \boldsymbol{\phi}(s,a))$ and $f_0(\boldsymbol{\theta}_k; \boldsymbol{\phi}(s,a))$ with any $\boldsymbol{\theta}_k$. Both terms diminish as $m \to \infty$. Theorem 1 implies the convergence to the optimal value $(\boldsymbol{\theta}^*, \lambda^*)$ is bounded by the approximation error, which will diminish to zero as representation power of $f_0(\boldsymbol{\theta}_k; \boldsymbol{\phi}(s,a))$ increases when $m \to \infty$. Finally, we note that the right hand side (17) ends up in $\mathcal{O}(1/k^2) + \mathcal{O}(1/k^{2/3}) + c$, where $c$ is a constant and its value goes to 0 as $m \to \infty$. This indicates the error bounds of linearization with the original neural network functions are controlled by the overparameterization value of $m$. Need to mention that a constant step size will result in a non-vanishing accumulated error as in [15].

**Remark 2.** *A finite-time analysis of nonlinear 2TSA was presented in [39]. However, [39] required a stability condition that $\lim_{k \to \infty}(\boldsymbol{\theta}_k, \lambda_k) = (\boldsymbol{\theta}^*, \lambda^*)$, and both h and g are locally approximated as linear functions. [19, 60] relaxed these conditions and provided a finite-time analysis under i.i.d. noise. These results were later extended to Markovian noise [20] under the assumption that H function is strongly monotone in $\boldsymbol{\theta}$ and G function is strongly monotone in $\lambda$. Since [20] leveraged the techniques in [19], it needed to explicitly characterize the covariance between the error caused by Markovian noise and the parameters' residual error in (14), leading to the convergence analysis much more intrinsic. [15] exploited the mixing time to avoid the covariance between the error caused by Markovian noise and the parameters' residual error, however, it only considered the single timescale Q-learning with linear function approximation. Though our* `Neural-Q-Whittle` *can be rewritten as a nonlinear 2TSA, the nonlinear parameterization of Q-function caused by the neural network function approximation makes the aforementioned analysis not directly applicable to ours and requires additional characterization as highlighted in Figure 1. The explicit characterization of approximation errors further distinguish our work.*

### 4.3 Proof Sketch

In this section, we sketch the proofs of the three steps shown in Figure 1 as required for Theorem 1.

#### 4.3.1 Step 1: Approximated Solution of `Neural-Q-Whittle`

We first approximate the optimal solution by projecting the Q-function in (6) to some function classes parameterized by $\boldsymbol{\theta}$. The common choice of the projected function classes is the local linearization of $f(\boldsymbol{\theta}; \boldsymbol{\phi}(s,a))$ at the initial point $\boldsymbol{\theta}_0$ [13, 62], i.e., $\mathcal{F} := \{f_0(\boldsymbol{\theta}; \boldsymbol{\phi}(s,a)), \forall \boldsymbol{\theta} \in \Theta\}$, where

$$f_0(\boldsymbol{\theta}; \boldsymbol{\phi}(s,a)) = \frac{1}{\sqrt{m}} \sum_{r=1}^{m} b_r \mathbb{1}\{\mathbf{w}_{r,0}^\mathsf{T} \boldsymbol{\phi}(s,a) > 0\} \mathbf{w}_r^\mathsf{T} \boldsymbol{\phi}(s,a). \tag{18}$$

Then, we define the approximate stationary point $\boldsymbol{\theta}_0^*$ with respect to $f_0(\boldsymbol{\theta}; \boldsymbol{\phi}(s,a))$ as follows.

**Definition 1.** *[[13, 62]] A point $\boldsymbol{\theta}_0^* \in \Theta$ is said to be the approximate stationary point of Algorithm 1 if for all feasible $\boldsymbol{\theta} \in \Theta$ it holds that $\mathbb{E}_{\mu,\pi,\mathcal{P}}[(\Delta_0 \cdot \nabla_{\boldsymbol{\theta}} f_0(\boldsymbol{\theta}; \boldsymbol{\phi}(s,a)))^\mathsf{T} (\boldsymbol{\theta} - \boldsymbol{\theta}_0^*)] \geq 0, \forall \boldsymbol{\theta} \in \Theta$, with $\Delta_0 := [r(s,a) + (1-a)\lambda^* - I_0(\boldsymbol{\theta}) + \max_{a'} f_0(\boldsymbol{\theta}; \boldsymbol{\phi}(s',a)) - f_0(\boldsymbol{\theta}; \boldsymbol{\phi}(s,a))]$, where $I_0(\boldsymbol{\theta}) = \frac{1}{2S} \sum_{s \in \mathcal{S}} [f_0(\boldsymbol{\theta}; \boldsymbol{\phi}(s,0)) + f_0(\boldsymbol{\theta}; \boldsymbol{\phi}(s,1))]$.*

Though there is a gap between the true neural function (6) and the approximated local linearized function (18), the gap diminishes as the width of neural network i.e., $m$, becomes large [13, 62].

With the approximated stationary point $\boldsymbol{\theta}_0^*$, we can redefine the two error terms in (14) as

$$\hat{\boldsymbol{\theta}}_k = \boldsymbol{\theta}_k - \boldsymbol{\theta}_0^*, \qquad \hat{\lambda}_k = \lambda_k - y_0(\boldsymbol{\theta}_k), \tag{19}$$

using which we correspondingly define a modified Lyapunov function $\hat{M}(\boldsymbol{\theta}_k, \lambda_k)$ in (16), where

$$y_0(\boldsymbol{\theta}) = r(s,1) + \sum_{s'} p(s'|s, 1) \max_a f_0(\boldsymbol{\theta}; \boldsymbol{\phi}(s', a)) - r(s,0) - \sum_{s'} p(s'|s, 0) \max_a f_0(\boldsymbol{\theta}; \boldsymbol{\phi}(s', a)). \tag{20}$$

### 4.3.2 Step 2: Convergence Rate of $\hat{M}(\boldsymbol{\theta}_k, \lambda_k)$ in (16)

Since we approximate the true neural network function $f(\boldsymbol{\theta}; \boldsymbol{\phi}(s, a))$ in (6) with the local linearized function $f_0(\boldsymbol{\theta}; \boldsymbol{\phi}(s, a))$ in (18), the operators $h(\cdot)$ and $g(\cdot)$ in (10)-(11) turn correspondingly to be

$$h_0(X_k, \boldsymbol{\theta}_k, \lambda_k) = \nabla_{\boldsymbol{\theta}} f_0(\boldsymbol{\theta}_k; \boldsymbol{\phi}(S_k, A_k)) \Delta_{k,0}, \; g_0(\boldsymbol{\theta}_k) := f_0(\boldsymbol{\theta}_k; \boldsymbol{\phi}(s, 1)) - f_0(\boldsymbol{\theta}_k; \boldsymbol{\phi}(s, 0)), \tag{21}$$

with $\Delta_{k,0} := r(S_k, A_k) + (1 - A_k)\lambda_k - I_0(\boldsymbol{\theta}_k) + \max_a f_0(\boldsymbol{\theta}_k; \boldsymbol{\phi}(S_{k+1}, a)) - f_0(\boldsymbol{\theta}_k; \boldsymbol{\phi}(S_k, A_k))$.

Before we present the finite-time error bound of the nonlinear 2TSA (9) under Markovian noise, we first discuss the mixing time of the Markov chain $\{X_k\}$ and our assumptions.

**Definition 2** (Mixing time [15]). *For any $\delta > 0$, define $\tau_\delta$ as*

$$\tau_\delta = \min\{k \geq 1 : \|\mathbb{E}[h_0(X_k, \boldsymbol{\theta}, \lambda)|X_0 = x] - H_0(\boldsymbol{\theta}, \lambda)\| \leq \delta(\|\boldsymbol{\theta} - \boldsymbol{\theta}_0^*\| + \|\lambda - y_0(\boldsymbol{\theta}_0^*)\|)\}. \tag{22}$$

**Assumption 1.** *The Markov chain $\{X_k\}$ is irreducible and aperiodic. Hence, there exists a unique stationary distribution $\mu$ [32], and constants $C > 0$ and $\rho \in (0, 1)$ such that $d_{TV}(P(X_k|X_0 = x), \mu) \leq C\rho^k, \forall k \geq 0, x \in \mathcal{X}$, where $d_{TV}(\cdot, \cdot)$ is the total-variation (TV) distance [32].*

**Remark 3.** *Assumption 1 is often assumed to study the asymptotic convergence of stochastic approximation under Markovian noise [4, 9, 15].*

**Lemma 1.** *The function $h_0(X, \boldsymbol{\theta}, \lambda)$ defined in (21) is globally Lipschitz continuous w.r.t $\boldsymbol{\theta}$ and $\lambda$ uniformly in $X$, i.e., $\|h_0(X, \boldsymbol{\theta}_1, \lambda_1) - h_0(X, \boldsymbol{\theta}_2, \lambda_2)\| \leq L_{h,1}\|\boldsymbol{\theta}_1 - \boldsymbol{\theta}_2\| + L_{h,2}\|\lambda_1 - \lambda_2\|, \forall X \in \mathcal{X}$, and $L_{h,1} = 3, h_{h,2} = 1$ are valid Lipschitz constants.*

**Lemma 2.** *The function $g_0(\boldsymbol{\theta})$ defined in (21) is linear and thus Lipschitz continuous in $\boldsymbol{\theta}$, i.e., $\|g_0(\boldsymbol{\theta}_1) - g_0(\boldsymbol{\theta}_2)\| \leq L_g\|\boldsymbol{\theta}_1 - \boldsymbol{\theta}_2\|$, and $L_g = 2$ is a valid Lipschitz constant.*

**Lemma 3.** *The function $y_0(\boldsymbol{\theta})$ defined in (20) is linear and thus Lipschitz continuous in $\boldsymbol{\theta}$, i.e., $\|y_0(\boldsymbol{\theta}_1) - y_0(\boldsymbol{\theta}_2)\| \leq L_y\|\boldsymbol{\theta}_1 - \boldsymbol{\theta}_2\|$, and $L_y = 2$ is a valid Lipschitz constant.*

**Remark 4.** *The Lipschitz continuity of $h_0$ guarantees the existence of a solution $\boldsymbol{\theta}$ to the ODE $\dot{\boldsymbol{\theta}}$ for a fixed $\lambda$, while the Lipschitz continuity of $g_0$ and $y_0$ ensures the existence of a solution $\lambda$ to the ODE $\dot{\lambda}$ when $\boldsymbol{\theta}$ is fixed. These lemmas often serve as assumptions when proving the convergence rate for both linear and nonlinear 2TSA [31, 39, 18, 24, 19, 17, 27].*

**Lemma 4.** *For a fixed $\lambda$, there exists a constant $\mu_1 > 0$ such that $h_0(X, \boldsymbol{\theta}, \lambda)$ defined in (10) satisfies*

$$\mathbb{E}[\hat{\boldsymbol{\theta}}^\mathsf{T} h_0(X, \boldsymbol{\theta}, \lambda)] \leq -\mu_1\|\hat{\boldsymbol{\theta}}\|^2.$$

*For fixed $\boldsymbol{\theta}$, there exists a constant $\mu_2 > 0$ such that $g_0(X, \boldsymbol{\theta}, \lambda)$ defined in (11) satisfies*

$$\mathbb{E}[\hat{\lambda} g_0(X, \boldsymbol{\theta}, \lambda)] \leq -\mu_2\|\hat{\lambda}\|^2.$$

**Remark 5.** *Lemma 4 guarantees the stability and uniqueness of the solution $\boldsymbol{\theta}$ to the ODE $\dot{\boldsymbol{\theta}}$ for a fixed $\lambda$, and the uniqueness of the solution $\lambda$ to the ODE $\dot{\lambda}$ for a fixed $\boldsymbol{\theta}$. This assumption can be viewed as a relaxation of the stronger monotone property of nonlinear mappings [19, 15], since it is automatically satisfied if $h$ and $g$ are strong monotone as assumed in [19].*

**Lemma 5.** *Under Assumption 1 and Lemma 1, there exist constants $C > 0$, $\rho \in (0, 1)$ and $L = \max(3, \max_X h_0(X, \boldsymbol{\theta}_0^*), y_0(\boldsymbol{\theta}_0^*))$ such that*

$$\tau_\delta \leq \frac{\log(1/\delta) + \log(2LCmd)}{\log(1/\rho)}.$$

**Remark 6.** $\tau_\delta$ *is equivalent to the mixing time of the underlying Markov chain satisfying* $\lim_{\delta\to 0}\delta\tau_\delta = 0$ *[15]. For simplicity, we remove the subscript and denote it as* $\tau$.

We now present the finite-time error bound for the Lyapunov function $\hat{M}(\boldsymbol{\theta}_k, \lambda_k)$ in (16).

**Theorem 2.** *Consider iterates* $\{\boldsymbol{\theta}_k\}$ *and* $\{\lambda_k\}$ *generated by* `Neural-Q-Whittle` *in (7) and (8). Given Lemma 1-4,* $\alpha_k = \frac{\alpha_0}{(k+1)}, \eta_k = \frac{\eta_0}{(k+1)^{4/3}}$, $C_1 := c_1(\|\boldsymbol{\theta}_0\| + \|\lambda_0\| + 1)$ *with a constant* $c_1$,

$$\mathbb{E}[\hat{M}(\boldsymbol{\theta}_{k+1}, \lambda_{k+1})|\mathcal{F}_{k-\tau}] \leq \frac{\tau^2 \mathbb{E}[\hat{M}(\boldsymbol{\theta}_\tau, \lambda_\tau)]}{(k+1)^2} + \frac{600\alpha_0^3}{\eta_0}\frac{(C_1 + \|\hat{\boldsymbol{\theta}}_0\|)^2 + (2C_1 + \|\hat{\lambda}_0\|)^2}{(k+1)^{2/3}}$$
$$+ \frac{\mathcal{O}\Big(c_1^3(\|\boldsymbol{\theta}_0\| + |\lambda_0| + 1)^3 m^{-1/2}\Big)}{(k+1)^{2/3}}, \quad \forall k \geq \tau. \quad (23)$$

### 4.3.3 Step 3: Approximation Error between $M(\boldsymbol{\theta}_k, \lambda_k)$ and $\hat{M}(\boldsymbol{\theta}_k, \lambda_k)$

Finally, we characterize the approximation error between Lyapunov functions $M(\boldsymbol{\theta}_k, \lambda_k)$ and $\hat{M}(\boldsymbol{\theta}_k, \lambda_k)$. Since we are dealing with long-term average MDP, we assume that the total variation of the MDP is bounded [47].

**Assumption 2.** *There exists* $0 < \kappa < 1$ *such that* $\sup_{(s,a),(s',a')}\|p(\cdot|s,a) - p(\cdot|s',a')\|_{TV} = 2\kappa$.

Hence, the Bellman operator is a span-contraction operator [47], i.e.,

$$span(\mathcal{T}f_0(\boldsymbol{\theta}_0^*) - \mathcal{T}f(\boldsymbol{\theta}^*)) \leq \kappa\, span(f_0(\boldsymbol{\theta}_0^*) - f(\boldsymbol{\theta}^*)). \quad (24)$$

**Assumption 3.** $\|\boldsymbol{\theta}_0^* - \boldsymbol{\theta}^*\| \leq c_0\|span(f_0(\boldsymbol{\theta}_0^*) - f(\boldsymbol{\theta}^*))\|$, *with* $c_0$ *being a positive constant.*

**Lemma 6.** *For* $M(\boldsymbol{\theta}_k, \lambda_k)$ *in (15) and* $\hat{M}(\boldsymbol{\theta}_k, \lambda_k)$ *in (16), with constants* $c_1$ *and* $c_0$ *(Assumption 3),*

$$M(\boldsymbol{\theta}_k, \lambda_k) \leq 2\hat{M}(\boldsymbol{\theta}_k, \lambda_k) + \frac{2\eta_k c_0^2}{\alpha_k(1-\kappa)}\|span(\Pi_\mathcal{F}f(\boldsymbol{\theta}^*) - f(\boldsymbol{\theta}^*))\| + 2\mathcal{O}\Big(\frac{c_1^3(\|\boldsymbol{\theta}_0\| + |\lambda_0| + 1)^3}{m^{1/2}}\Big).$$

## 5 Numerical Experiments

We numerically evaluate the performance of `Neural-Q-Whittle` using an example of circulant dynamics [23, 3, 8]. The state space is $\mathcal{S} = \{1, 2, 3, 4\}$. Rewards are $r(1, a) = -1, r(2, a) = r(3, a) = 0$, and $r(4, a) = 1$ for $a \in \{0, 1\}$. The dynamics of states are circulant and defined as

$$P^1 = \begin{bmatrix} 0.5 & 0.5 & 0 & 0 \\ 0 & 0.5 & 0.5 & 0 \\ 0 & 0 & 0.5 & 0.5 \\ 0.5 & 0 & 0 & 0.5 \end{bmatrix} \text{ and } P^0 = \begin{bmatrix} 0.5 & 0 & 0 & 0.5 \\ 0.5 & 0.5 & 0 & 0 \\ 0 & 0.5 & 0.5 & 0 \\ 0 & 0 & 0.5 & 0.5 \end{bmatrix}.$$

This indicates that the process either remains in its current state or increments if it is active (i.e., $a = 1$), or it either remains the current state or decrements if it is passive (i.e., $a = 0$). The exact value of Whittle indices [23] are $\lambda(1) = -0.5, \lambda(2) = 0.5, \lambda(3) = 1$, and $\lambda(4) = -1$.

In our experiments, we set the learning rates as $\alpha_k = 0.5/(k+1)$ and $\eta_k = 0.1/(k+1)^{4/3}$. We use $\epsilon$-greedy for the exploration and exploitation tradeoff with $\epsilon = 0.5$. We consider a two-layer neural network with the number of neurons in the hidden layer as $m = 200$. As described in Algorithm 1, $b_r, \forall r$ are uniformly initialized in $\{-1, 1\}$ and $w_r, \forall r$ are initialized as a zero mean Gaussian distribution according to $\mathcal{N}(\mathbf{0}, \mathbf{I}_d/d)$. These results are carried out by Monte Carlo simulations with 100 independent trials.

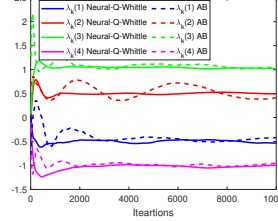

(a) `Neural-Q-Whittle` vs. `Q-Whittle` [3].

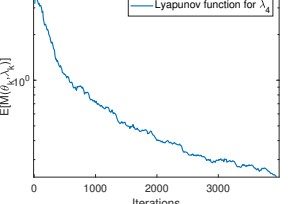

(b) Convergence of Lyapunov function in (15).

Figure 2: Convergence of `Neural-Q-Whittle`.

**Convergence to true Whittle index.** First, we verify that `Neural-Q-Whittle` convergences to true Whittle indices, and compare to `Q-Whittle`, the first Whittle index based Q-learning algorithm. As illustrated in Figure 2a, `Neural-Q-Whittle` guarantees the convergence to true Whittle indices and outperforms `Q-Whittle` [3] in the convergence speed. This is due to the fact that `Neural-Q-Whittle` updates the Whittle index of a specific state even when the current visited state is not that state.

Second, we further compare with other other Whittle index learning algorithms, i.e., `Q-Whittle-LFA` [60], WIQL [8] and QWIC [23]in Figure 3. As we observe from Figure 3, only `Neural-Q-Whittle` and `Q-Whittle-LFA` in [60] can converge to the true Whittle indices for each state, while the other two benchmarks algorithms do not guarantee the convergence of true Whittle indices. Interestingly, the learning Whittle indices converge and maintain a correct relative order of magnitude, which is still be able to be used in real world problems [60]. Moreover, we observe that `Neural-Q-Whittle` achieves similar convergence performance as `Q-Whittle-LFA` in the considered example, whereas the latter has been shown to achieve good performance in real world applications in

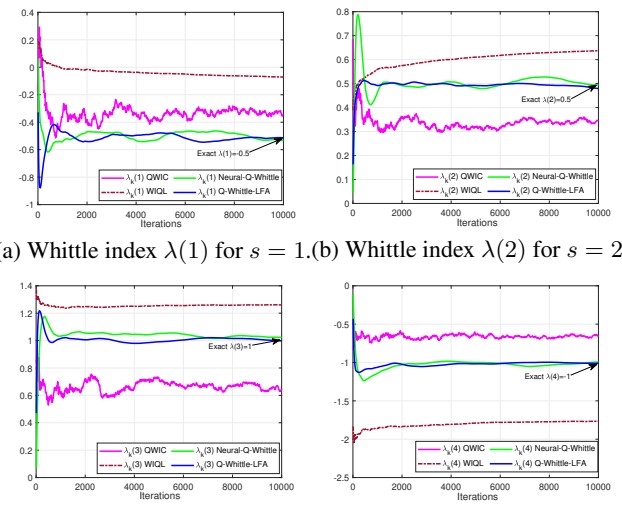

(a) Whittle index $\lambda(1)$ for $s = 1$. (b) Whittle index $\lambda(2)$ for $s = 2$.

(c) Whittle index $\lambda(3)$ $s = 3$. (d) Whittle index $\lambda(4)$ for $s = 4$.

Figure 3: Convergence comparison between `Neural-Q-Whittle` and benchmark algorithms.

[60]. Though this work focuses on the theoretical convergence analysis of Q-learning based whittle index under the neural network function approximation, it might be promising to implement it in real-world applications to fully leverage the strong representation ability of neural network functions, which serves as future investigation of this work.

**Convergence of the Lyapunov function defined in (15).** We also evaluate the convergence of the proposed Lyapunov function defined in (15), which is presented in Figure 2b. It depicts $\mathbb{E}[M(\boldsymbol{\theta}_k, \lambda_k)]$ vs. the number of iterations in logarithmic scale. For ease of presentation, we only take state $s = 4$ as an illustrative example. It is clear that $M(\boldsymbol{\theta}_k, \lambda_k)$ converges to zero as the number of iterations increases, which is in alignment with our theoretical results in Theorem 1.

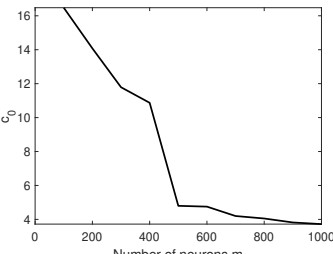

Figure 4: Verification of Assumption 3 w.r.t the constant $c_0$.

**Verification of Assumption 3.** We now verify Assumption 3 that the gap between $\boldsymbol{\theta}_0^*$ and $\boldsymbol{\theta}^*$ can be bounded by the span of $f_0(\boldsymbol{\theta}_0^*)$ and $f(\boldsymbol{\theta}^*)$ with a constant $c_0$. In Figure 4, we show $c_0$ as a function of the number of neurons in the hidden layer $m$. It clearly indicates that constant $c_0$ exists and decreases as the number of neurons grows larger.

## 6 Conclusion

We presented `Neural-Q-Whittle`, a Whittle index based Q-learning algorithm for RMAB with neural network function approximation. We proved that `Neural-Q-Whittle` achieves an $\mathcal{O}(1/k^{2/3})$ convergence rate, where $k$ is the number of iterations when data are generated from a Markov chain and Q-function is approximated by a ReLU neural network. By viewing `Neural-Q-Whittle` as 2TSA and leveraging the Lyapunov drift method, we removed the projection step on parameter update of Q-learning with neural network function approximation. Extending the current framework to two-timescale Q-learning (i.e., the coupled iterates between Q-function values and Whittle indices) with general deep neural network approximation is our future work.

## Acknowledgements

This work was supported in part by the National Science Foundation (NSF) grant RINGS-2148309, and was supported in part by funds from OUSD R&E, NIST, and industry partners as specified in the Resilient & Intelligent NextG Systems (RINGS) program. This work was also supported in part by the U.S. Army Research Office (ARO) grant W911NF-23-1-0072, and the U.S. Department of Energy (DOE) grant DE-EE0009341. Any opinions, findings, and conclusions or recommendations expressed in this material are those of the authors and do not necessarily reflect the views of the funding agencies.

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

# A   Related Work

**Online Restless Bandits.** The online RMAB setting, where the underlying MDPs are unknown, has been gaining attention, e.g., [16, 33, 34, 50, 42, 26]. However, these methods do not exploit the special structure available in the problem and contend directly with an extremely high dimensional state-action space yielding the algorithms to be too slow to be useful. Recently, RL based algorithms have been developed [10, 23, 53, 8, 28, 57, 59, 61, 58, 3], to explore the problem structure through index policies. For instance, [23] proposed a Q-learning algorithm for Whittle index under the discounted setting, which lacks of convergence guarantees. [8] approximated Whittle index using the difference of $Q(s, 1) - Q(s, 0)$ for any state $s$, which is not guaranteed to converge to the true Whittle index in general scenarios. To our best knowledge, the `Q-Whittle` in (4)-(5) proposed by [3] is the first algorithm with a rigorous asymptotic analysis. Therefore, [23, 3, 8, 28] lacked finite-time performance analysis and multi-timescale stochastic approximation algorithms usually suffer from slow convergence.

[53, 57] designed model-based low-complexity policy but is constrained to either a specific Markovian model or depends on a simulator for a finite-horizon setting which cannot be directly applied here. Latter on, [60] showed the finite-time convergence performance under the `Q-Whittle` setting of [3] with linear function approximation. However, the underlying assumption in [3, 60] is that data samples are drawn i.i.d per iteration. This is often not the case in practice since data samples of Q-learning are drawn according to the underlying Markov decision process. Till now, the finite-time convergence rate of `Q-Whittle` under the more challenging Markovian setting remains to be an open problem. Though [43] proposed a novel DQN method and applied it to Whittle index learning, it lacks of theoretical convergence analysis. To our best knowledge, our work is the first to study low-complexity model-free Q-learning for RMAB with neural network function approximation and provide a finite-time performance guarantee.

**Two-Timescale Stochastic Approximation.** The theoretical understanding of average-reward reinforcement learning (RL) methods is limited. Most existing results focus on asymptotic convergence [51, 1, 52, 65], or finite-time performance guarantee for discounted Q-learning [15, 46, 14]. However, the analysis of average-reward RL algorithms is known to be more challenging than their discounted-reward counterparts [66, 55]. In particular, our `Neural-Q-Whittle` follows the 2TSA scheme [11, 30, 7]. The standard technique for analyzing 2TSA is via the ODE method to prove asymptotic convergence [9]. Building off the importance of asymptotic results, recent years have witnessed a focus shifted to non-asymptotic, finite-time analysis of 2TSA [24, 21, 19, 63]. The closest work is [19], which characterized the convergence rate for a general non-linear 2TSA with i.i.d. noise. We generalize this result to provide a finite-time analysis of our `Neural-Q-Whittle` with Markovian noise. In addition, existing finite-time analysis, e.g., sample complexity [66] and regret [55] of Q-learning with average reward focus on a single-timescale SA, and hence cannot be directly applied to our `Neural-Q-Whittle`. Finally, existing Q-learning with linear function approximation [37, 6, 67] and neural network function approximation [13, 62] requires an additional projection step onto a bounded set related to the unknown stationary distribution of the underlying MDPs, or focuses on a single-timescale SA [15].

# B   Review on Whittle Index Policy

Whittle index policy addresses the intractable issue of RMAB through decomposition. In each round $t$, it first calculates the Whittle index for each arm $n$ independently only based on its current state $s_n(t)$, and then the Whittle index policy simply selects the $K$ arms with the highest indices to activate. Following Whittle's approach[56], we can consider a system with only one arm due to the decomposition, and the Lagrangian is expressed as

$$L(\pi, \lambda) = \liminf_{T \to \infty} \frac{1}{T} \mathbb{E}_\pi \sum_{t=1}^{T} \left\{ r(t) + \lambda \left( 1 - a(t) \right) \right\}, \tag{25}$$

where $\lambda$ is the Lagrangian multiplier (or the subsidy for selecting passive action). For a particular $\lambda$, the optimal activation policy can be expressed by a set of states in which it would activate this arm, which is denoted $D(\lambda)$.

**Definition 3** (Indexability). *We denote $D(\lambda)$ as the set of states $S$ for which the optimal action for the arm is to choose a passive action, i.e., $A = 0$. Then the arm is said to be indexable if $D(\lambda)$ increases with $\lambda$, i.e., if $\lambda > \lambda'$, then $D(\lambda) \supseteq D(\lambda')$.*

Following the indexability property, the Whittle index in a particular state $S$ is defined as follows.

**Definition 4** (Whittle Index). *The Whittle index in state $S$ for the indexable arm is the smallest value of the Lagrangian multiplier $\lambda$ such that the optimal policy at state $S$ is indifferent towards actions $A = 0$ and $A = 1$. We denote such a Whittle index as $\lambda(S)$ satisfying $\lambda(S) := \inf_{\lambda \geq 0}\{S \in D(\lambda)\}$.*

**Definition 5** (Whittle index policy). *Whittle index policy is a controlled policy which activates the $K$ arms with the highest whittle index $\lambda_i(S_i(t))$ at each time slot $t$.*

# C Proof of Lemmas for "Step 2: Convergence Rate of $\hat{M}(\boldsymbol{\theta}_k, \lambda_k)$ in (16)"

## C.1 Proof of Lemma 1

*Proof.* Recall that

$$f_0(\boldsymbol{\theta}; \boldsymbol{\phi}(s, a)) = \frac{1}{\sqrt{m}} \sum_{r=1}^{m} b_r \mathbb{1}\{\mathbf{w}_{r,0}^{\mathsf{T}} \boldsymbol{\phi}(s, a) > 0\} \mathbf{w}_r^{\mathsf{T}} \boldsymbol{\phi}(s, a).$$

Thus we denote $\nabla_{\boldsymbol{\theta}} f_0(\boldsymbol{\theta}; \boldsymbol{\phi}(s, a))$ as

$$\nabla_{\boldsymbol{\theta}} f_0(\boldsymbol{\theta}; \boldsymbol{\phi}(s, a)) := \Big[ \frac{1}{\sqrt{m}} b_1 \mathbb{1}\{\mathbf{w}_{1,0}^{\mathsf{T}} \boldsymbol{\phi}(s, a) > 0\} \boldsymbol{\phi}(s, a)^{\mathsf{T}}, \dots,$$

$$\frac{1}{\sqrt{m}} b_m \mathbb{1}\{\mathbf{w}_{m,0}^{\mathsf{T}} \boldsymbol{\phi}(s, a) > 0\} \boldsymbol{\phi}(s, a)^{\mathsf{T}} \Big]^{\mathsf{T}}. \tag{26}$$

Since $\|\boldsymbol{\phi}(s, a)\| \leq 1, \forall s \in \mathcal{S}, a \in \mathcal{A}$ and the fact that $b_r, \forall r \in [m]$ is uniformly initialized as 1 and $-1$, we have $\|\nabla_{\boldsymbol{\theta}} f_0(\boldsymbol{\theta}; \boldsymbol{\phi}(s, a))\| \leq 1$.

Therefore, we have the following inequality for any parameter pairs $(\boldsymbol{\theta}_1, \lambda_1)$ and $(\boldsymbol{\theta}_2, \lambda_2)$ with $X = (s, a, s') \in \mathcal{X}$,

$$\|h_0(X, \boldsymbol{\theta}_1, \lambda_1) - h_0(X, \boldsymbol{\theta}_2, \lambda_2)\|$$

$$= \Big\| \nabla_{\boldsymbol{\theta}} f_0(\boldsymbol{\theta}_1; \boldsymbol{\phi}(s, a)) \Big[ r(s, a) + (1 - a)\lambda_1 - I_0(\boldsymbol{\theta}_1) + \max_{a_1} f_0(\boldsymbol{\theta}_1; \boldsymbol{\phi}(s', a_1)) - f_0(\boldsymbol{\theta}_1; \boldsymbol{\phi}(s, a)) \Big]$$

$$- \nabla_{\boldsymbol{\theta}} f_0(\boldsymbol{\theta}_2; \boldsymbol{\phi}(s, a)) \Big[ r(s, a) + (1 - a)\lambda_2 - I_0(\boldsymbol{\theta}_2) + \max_{a_2} f_0(\boldsymbol{\theta}_2; \boldsymbol{\phi}(s', a_2)) - f_0(\boldsymbol{\theta}_2; \boldsymbol{\phi}(s, a)) \Big] \Big\|$$

$$\overset{(a_1)}{=} \Big\| \nabla_{\boldsymbol{\theta}} f_0(\boldsymbol{\theta}_1; \boldsymbol{\phi}(s, a)) \Big[ (1 - a)(\lambda_1 - \lambda_2) + I_0(\boldsymbol{\theta}_2) - I_0(\boldsymbol{\theta}_1) + f_0(\boldsymbol{\theta}_2; \boldsymbol{\phi}(s, a)) - f_0(\boldsymbol{\theta}_1; \boldsymbol{\phi}(s, a))$$

$$+ \max_{a_1} \Big( f_0(\boldsymbol{\theta}_1; \boldsymbol{\phi}(s', a_1)) - \max_{a_2} f_0(\boldsymbol{\theta}_2; \boldsymbol{\phi}(s', a_2)) \Big) \Big] \Big\|$$

$$\overset{(a_2)}{\leq} \|(1 - a)(\lambda_1 - \lambda_2)\| + \|f_0(\boldsymbol{\theta}_2; \boldsymbol{\phi}(s, a)) - f_0(\boldsymbol{\theta}_1; \boldsymbol{\phi}(s, a))\|$$

$$+ \Big\| \frac{1}{2S} \sum_{\tilde{s} \in \mathcal{S}} f_0(\boldsymbol{\theta}_2; \boldsymbol{\phi}(\tilde{s}, 0)) - f_0(\boldsymbol{\theta}_1; \boldsymbol{\phi}(\tilde{s}, 0)) + f_0(\boldsymbol{\theta}_2; \boldsymbol{\phi}(\tilde{s}, 1)) - f_0(\boldsymbol{\theta}_1; \boldsymbol{\phi}(\tilde{s}, 1)) \Big\|$$

$$+ \Big\| \max_{a_1} \Big( f_0(\boldsymbol{\theta}_1; \boldsymbol{\phi}(s', a_1)) - \max_{a_2} f_0(\boldsymbol{\theta}_2; \boldsymbol{\phi}(s', a_2)) \Big) \Big\|$$

$$\overset{(a_3)}{\leq} \|(1 - a)(\lambda_1 - \lambda_2)\| + \|\nabla_{\boldsymbol{\theta}} f_0(\boldsymbol{\theta}_1; \boldsymbol{\phi}(s, a))(\boldsymbol{\theta}_2 - \boldsymbol{\theta}_1)\|$$

$$+ \Big\| \frac{1}{2S} \sum_{\tilde{s} \in \mathcal{S}} \nabla_{\boldsymbol{\theta}} f_0(\boldsymbol{\theta}_1; \boldsymbol{\phi}(\tilde{s}, 0))(\boldsymbol{\theta}_2 - \boldsymbol{\theta}_1) + \nabla_{\boldsymbol{\theta}} f_0(\boldsymbol{\theta}_1; \boldsymbol{\phi}(\tilde{s}, 1))(\boldsymbol{\theta}_2 - \boldsymbol{\theta}_1) \Big\|$$

$$+ \Big\| \max_{a_1} \Big( f_0(\boldsymbol{\theta}_1; \boldsymbol{\phi}(s', a_1)) - \max_{a_2} f_0(\boldsymbol{\theta}_2; \boldsymbol{\phi}(s', a_2)) \Big) \Big\|$$

$$\overset{(a_4)}{\leq} \|(\lambda_1 - \lambda_2)\| + 2\|\boldsymbol{\theta}_1 - \boldsymbol{\theta}_2\| + \Big\| \max_{a_1} \Big( f_0(\boldsymbol{\theta}_1; \boldsymbol{\phi}(s', a_1)) - \max_{a_2} f_0(\boldsymbol{\theta}_2; \boldsymbol{\phi}(s', a_2)) \Big) \Big\|$$

$$\overset{(a_5)}{\leq} \|(\lambda_1 - \lambda_2)\| + 2\|\boldsymbol{\theta}_1 - \boldsymbol{\theta}_2\| + \left\| \max_{a'} f_0(\boldsymbol{\theta}_1; \boldsymbol{\phi}(s', a')) - f_0(\boldsymbol{\theta}_2; \boldsymbol{\phi}(s', a')) \right\|$$

$$\overset{(a_6)}{\leq} \|(\lambda_1 - \lambda_2)\| + 3\|\boldsymbol{\theta}_1 - \boldsymbol{\theta}_2\|.$$

Specifically, $(a_1)$ holds due to the fact that $\nabla_{\boldsymbol{\theta}} f_0(\boldsymbol{\theta}_1; \boldsymbol{\phi}(s, a)) = \nabla_{\boldsymbol{\theta}} f_0(\boldsymbol{\theta}_2; \boldsymbol{\phi}(s, a))$ as in (26). Since

$$I_0(\boldsymbol{\theta}_k) = \frac{1}{2S} \sum_{\tilde{s} \in \mathcal{S}} \Big[ f_0(\boldsymbol{\theta}_k; \boldsymbol{\phi}(\tilde{s}, 0)) + f_0(\boldsymbol{\theta}_k; \boldsymbol{\phi}(\tilde{s}, 1)) \Big],$$

$(a_2)$ is due to the fact that $\|\boldsymbol{x} + \boldsymbol{y}\| \leq \|\boldsymbol{x}\| + \|\boldsymbol{y}\|, \forall \boldsymbol{x}, \boldsymbol{y} \in \mathbb{R}^{md}$ and $\|\boldsymbol{x} \cdot \boldsymbol{y}\| \leq \|\boldsymbol{x}\| \cdot \|\boldsymbol{y}\|, \forall \boldsymbol{x}, \boldsymbol{y} \in \mathbb{R}^{md}$ and $\|\boldsymbol{\phi}(s, a)\| \leq 1, \forall s, a$. $(a_3)$ holds since

$$f_0(\boldsymbol{\theta}_2; \boldsymbol{\phi}(s, a)) - f_0(\boldsymbol{\theta}_1; \boldsymbol{\phi}(s, a)) = \nabla_{\boldsymbol{\theta}} f_0(\boldsymbol{\theta}_1; \boldsymbol{\phi}(s, a))(\boldsymbol{\theta}_2 - \boldsymbol{\theta}_1), \forall s \in \mathcal{S}, a \in \mathcal{A}. \tag{27}$$

$(a_4)$ holds for the same reason as $(a_2)$. $(a_5)$ is due to the fact that

$$\| \max_{a'} f_0(\boldsymbol{\theta}_1; \boldsymbol{\phi}(s', a')) - f_0(\boldsymbol{\theta}_2; \boldsymbol{\phi}(s', a'))\| \leq \max \Bigg( \left\| \max_{a'} f_0(\boldsymbol{\theta}_1; \boldsymbol{\phi}(s', a')) - f_0(\boldsymbol{\theta}_2; \boldsymbol{\phi}(s', a')) \right\|,$$

$$\left\| \min_{a'} f_0(\boldsymbol{\theta}_1; \boldsymbol{\phi}(s', a')) - f_0(\boldsymbol{\theta}_2; \boldsymbol{\phi}(s', a')) \right\| \Bigg). \tag{28}$$

$(a_6)$ holds for the same reason as $(a_3)$ and $(a_4)$. $\qquad \square$

## C.2 Proof of Lemma 2

*Proof.* Since $g_0(\cdot)$ is irrelevant with $X$ and $\lambda$, in the following, we write $g_0(X, \boldsymbol{\theta}, \lambda)$ with $g_0(\lambda)$ interchangeably. For any $\boldsymbol{\theta}_1 \in \mathbb{R}^{md}$ and $\boldsymbol{\theta}_2 \in \mathbb{R}^{md}$, we have

$$\|g_0(\boldsymbol{\theta}_1) - g_0(\boldsymbol{\theta}_2)\|$$
$$= \|f_0(\boldsymbol{\theta}_1; \boldsymbol{\phi}(s, 1)) - f_0(\boldsymbol{\theta}_1; \boldsymbol{\phi}(s, 0)) - f_0(\boldsymbol{\theta}_2; \boldsymbol{\phi}(s, 1)) + f_0(\boldsymbol{\theta}_2; \boldsymbol{\phi}(s, 0))\|$$
$$\leq \|f_0(\boldsymbol{\theta}_1; \boldsymbol{\phi}(s, 1)) - f_0(\boldsymbol{\theta}_2; \boldsymbol{\phi}(s, 1))\| + \|f_0(\boldsymbol{\theta}_1; \boldsymbol{\phi}(s, 0)) - f_0(\boldsymbol{\theta}_2; \boldsymbol{\phi}(s, 0))\|$$
$$= \|\nabla_{\boldsymbol{\theta}} f_0(\boldsymbol{\theta}_1; \boldsymbol{\phi}(s, 1))(\boldsymbol{\theta}_2 - \boldsymbol{\theta}_1)\| + \|\nabla_{\boldsymbol{\theta}} f_0(\boldsymbol{\theta}_1; \boldsymbol{\phi}(s, 0))(\boldsymbol{\theta}_2 - \boldsymbol{\theta}_1)\|$$
$$\leq \|\nabla_{\boldsymbol{\theta}} f_0(\boldsymbol{\theta}_1; \boldsymbol{\phi}(s, 1))\| \cdot \|\boldsymbol{\theta}_1 - \boldsymbol{\theta}_2\| + \|\nabla_{\boldsymbol{\theta}} f_0(\boldsymbol{\theta}_1; \boldsymbol{\phi}(s, 0))\| \cdot \|\boldsymbol{\theta}_1 - \boldsymbol{\theta}_2\|$$
$$\leq 2\|\boldsymbol{\theta}_1 - \boldsymbol{\theta}_2\|,$$

where the first inequality is due to the fact that $\|\boldsymbol{x} + \boldsymbol{y}\| \leq \|\boldsymbol{x}\| + \|\boldsymbol{y}\|, \forall \boldsymbol{x}, \boldsymbol{y} \in \mathbb{R}^{md}$, the second inequality holds due to $\|\boldsymbol{x} \cdot \boldsymbol{y}\| \leq \|\boldsymbol{x}\| \cdot \|\boldsymbol{y}\|, \forall \boldsymbol{x}, \boldsymbol{y} \in \mathbb{R}^{md}$, and the last inequality holds since $\|\nabla_{\boldsymbol{\theta}} f_0(\boldsymbol{\theta}_1; \boldsymbol{\phi}(s, a))\| \leq 1, \forall s \in \mathcal{S}, a \in \mathcal{A}. \qquad \square$

## C.3 Proof of Lemma 3

*Proof.* For any $\boldsymbol{\theta}_1 \in \mathbb{R}^{md}$ and $\boldsymbol{\theta}_2 \in \mathbb{R}^{md}$, we have

$\|y_0(\boldsymbol{\theta}_1) - y_0(\boldsymbol{\theta}_2)\|$

$$= \Bigg\| r(s, 1) - r(s, 0) + \sum_{s'} P(s'|s, 1) \max_a f_0(\boldsymbol{\theta}_1; \boldsymbol{\phi}(s', a)) - \sum_{s'} P(s'|s, 0) \max_a f_0(\boldsymbol{\theta}_1; \boldsymbol{\phi}(s', a))$$

$$- r(s, 1) - r(s, 0) + \sum_{s'} P(s'|s, 1) \max_a f_0(\boldsymbol{\theta}_2; \boldsymbol{\phi}(s', a)) - \sum_{s'} P(s'|s, 0) \max_a f_0(\boldsymbol{\theta}_2; \boldsymbol{\phi}(s', a)) \Bigg\|$$

$$= \Bigg\| \sum_{s'} P(s'|s, 1) \max_a f_0(\boldsymbol{\theta}_1; \boldsymbol{\phi}(s', a)) - \sum_{s'} P(s'|s, 1) \max_a f_0(\boldsymbol{\theta}_2; \boldsymbol{\phi}(s', a))$$

$$- \sum_{s'} P(s'|s, 0) \max_a f_0(\boldsymbol{\theta}_1; \boldsymbol{\phi}(s', a)) + \sum_{s'} P(s'|s, 0) \max_a f_0(\boldsymbol{\theta}_2; \boldsymbol{\phi}(s', a)) \Bigg\|$$

$$\leq \Bigg\| \sum_{s'} P(s'|s, 1) \max_a f_0(\boldsymbol{\theta}_1; \boldsymbol{\phi}(s', a)) - \sum_{s'} P(s'|s, 1) \max_a f_0(\boldsymbol{\theta}_2; \boldsymbol{\phi}(s', a)) \Bigg\|$$

$$+ \Bigg\| \sum_{s'} P(s'|s, 0) \max_a f_0(\boldsymbol{\theta}_1; \boldsymbol{\phi}(s', a)) + \sum_{s'} P(s'|s, 0) \max_a f_0(\boldsymbol{\theta}_2; \boldsymbol{\phi}(s', a)) \Bigg\|$$

$$\leq 2\|\boldsymbol{\theta}_1 - \boldsymbol{\theta}_2\|,$$

with the last inequality holds due to (27) and (28). $\qquad \square$

## C.4 Proof of Lemma 4

*Proof.* 1) We first show that there exists a constant $\mu_1 > 0$ such that $\mathbb{E}[\hat{\boldsymbol{\theta}}^{\mathsf{T}} h_0(X, \boldsymbol{\theta}, \lambda)] \leq -\mu_1 \|\hat{\boldsymbol{\theta}}\|^2$. According to the definition of $\boldsymbol{\theta}_0^*$ given in Definition 1, $\mathbb{E}[h_0(X, \boldsymbol{\theta}_0^*, y_0(\boldsymbol{\theta}_0^*))] = 0$. Hence, we have

$$
\mathbb{E}\left[\hat{\boldsymbol{\theta}}^{\mathsf{T}}(h_0(X, \boldsymbol{\theta}, \lambda) - h_0(X, \boldsymbol{\theta}_0^*, y_0(\boldsymbol{\theta}_0^*)))\right]
$$

$$
= \hat{\boldsymbol{\theta}}^{\mathsf{T}} \mathbb{E}[h_0(X, \boldsymbol{\theta}, \lambda) - h_0(X, \boldsymbol{\theta}_0^*, y_0(\boldsymbol{\theta}_0^*))]
$$

$$
= \hat{\boldsymbol{\theta}}^{\mathsf{T}} \mathbb{E}\left[\nabla_{\boldsymbol{\theta}} f_0(\boldsymbol{\theta}; \boldsymbol{\phi}(s, a))\left[r(s, a) + (1-a)\lambda - I_0(\boldsymbol{\theta}) + \max_{a_1} f_0(\boldsymbol{\theta}; \boldsymbol{\phi}(s', a_1)) - f_0(\boldsymbol{\theta}; \boldsymbol{\phi}(s, a))\right]\right.
$$

$$
\left. - \nabla_{\boldsymbol{\theta}} f_0(\boldsymbol{\theta}_0^*; \boldsymbol{\phi}(s, a))\left[r(s, a) + (1-a)y_0(\boldsymbol{\theta}_0^*) - I_0(\boldsymbol{\theta}_0^*) + \max_{a_2} f_0(\boldsymbol{\theta}_0^*; \boldsymbol{\phi}(s', a_2)) - f_0(\boldsymbol{\theta}_0^*; \boldsymbol{\phi}(s, a))\right]\right]
$$

$$
\overset{(b_1)}{=} \hat{\boldsymbol{\theta}}^{\mathsf{T}} \mathbb{E}\left[\nabla_{\boldsymbol{\theta}} f_0(\boldsymbol{\theta}; \boldsymbol{\phi}(s, a))\left[(1-a)(\lambda - y_0(\boldsymbol{\theta}_0^*)) + I(\boldsymbol{\theta}_0^*) - I_0(\boldsymbol{\theta}) + f_0(\boldsymbol{\theta}_0^*; \boldsymbol{\phi}(s, a)) - f_0(\boldsymbol{\theta}; \boldsymbol{\phi}(s, a))\right.\right.
$$

$$
\left.\left. + \max_{a_1} f_0(\boldsymbol{\theta}; \boldsymbol{\phi}(s', a_1)) - \max_{a_2} f_0(\boldsymbol{\theta}_0^*; \boldsymbol{\phi}(s', a_2))\right]\right]
$$

$$
= \hat{\boldsymbol{\theta}}^{\mathsf{T}} \mathbb{E}\left[\nabla_{\boldsymbol{\theta}} f_0(\boldsymbol{\theta}; \boldsymbol{\phi}(s, a))\left[\max_{a_1} f_0(\boldsymbol{\theta}; \boldsymbol{\phi}(s', a_1)) - \max_{a_2} f_0(\boldsymbol{\theta}_0^*; \boldsymbol{\phi}(s', a_2))\right]\right]
$$

$$
- \hat{\boldsymbol{\theta}}^{\mathsf{T}} \mathbb{E}\left[\nabla_{\boldsymbol{\theta}} f_0(\boldsymbol{\theta}; \boldsymbol{\phi}(s, a))[I_0(\boldsymbol{\theta}) - I_0(\boldsymbol{\theta}_0^*)]\right] - \hat{\boldsymbol{\theta}}^{\mathsf{T}} \mathbb{E}\left[\nabla_{\boldsymbol{\theta}} f_0(\boldsymbol{\theta}; \boldsymbol{\phi}(s, a))[f_0(\boldsymbol{\theta}; \boldsymbol{\phi}(s, a)) - f_0(\boldsymbol{\theta}_0^*; \boldsymbol{\phi}(s, a))]\right]
$$

$$
+ \hat{\boldsymbol{\theta}}^{\mathsf{T}} \mathbb{E}\left[\nabla_{\boldsymbol{\theta}} f_0(\boldsymbol{\theta}; \boldsymbol{\phi}(s, a))[(1-a)(\lambda - y_0(\boldsymbol{\theta}_0^*))]\right]
$$

$$
\overset{(b_2)}{\leq} \hat{\boldsymbol{\theta}}^{\mathsf{T}} \mathbb{E}\left[\nabla_{\boldsymbol{\theta}} f_0(\boldsymbol{\theta}; \boldsymbol{\phi}(s, a)) \max_{a'}\left[f_0(\boldsymbol{\theta}; \boldsymbol{\phi}(s', a')) - f_0(\boldsymbol{\theta}_0^*; \boldsymbol{\phi}(s', a'))\right]\right]
$$

$$
- \hat{\boldsymbol{\theta}}^{\mathsf{T}} \mathbb{E}\left[\nabla_{\boldsymbol{\theta}} f_0(\boldsymbol{\theta}; \boldsymbol{\phi}(s, a))[I_0(\boldsymbol{\theta}) - I_0(\boldsymbol{\theta}_0^*)]\right] - \hat{\boldsymbol{\theta}}^{\mathsf{T}} \mathbb{E}\left[\nabla_{\boldsymbol{\theta}} f_0(\boldsymbol{\theta}; \boldsymbol{\phi}(s, a))[f_0(\boldsymbol{\theta}; \boldsymbol{\phi}(s, a)) - f_0(\boldsymbol{\theta}_0^*; \boldsymbol{\phi}(s, a))]\right]
$$

$$
+ \hat{\boldsymbol{\theta}}^{\mathsf{T}} \mathbb{E}\left[\nabla_{\boldsymbol{\theta}} f_0(\boldsymbol{\theta}; \boldsymbol{\phi}(s, a))[(1-a)(\lambda - y_0(\boldsymbol{\theta}_0^*))]\right]
$$

$$
\overset{(b_3)}{\leq} \hat{\boldsymbol{\theta}}^{\mathsf{T}} \mathbb{E}\left[\nabla_{\boldsymbol{\theta}} f_0(\boldsymbol{\theta}; \boldsymbol{\phi}(s, a)) \max_{a'}\left[f_0(\boldsymbol{\theta}; \boldsymbol{\phi}(s', a')) - f_0(\boldsymbol{\theta}_0^*; \boldsymbol{\phi}(s', a'))\right]\right]
$$

$$
- \hat{\boldsymbol{\theta}}^{\mathsf{T}} \mathbb{E}\left[\nabla_{\boldsymbol{\theta}} f_0(\boldsymbol{\theta}; \boldsymbol{\phi}(s, a))[I_0(\boldsymbol{\theta}) - I_0(\boldsymbol{\theta}_0^*)]\right] - \hat{\boldsymbol{\theta}}^{\mathsf{T}} \mathbb{E}\left[\nabla_{\boldsymbol{\theta}} f_0(\boldsymbol{\theta}; \boldsymbol{\phi}(s, a))[f_0(\boldsymbol{\theta}; \boldsymbol{\phi}(s, a)) - f_0(\boldsymbol{\theta}_0^*; \boldsymbol{\phi}(s, a))]\right]
$$

$$
\overset{(b_4)}{=} \|\hat{\boldsymbol{\theta}}\|^2 \mathbb{E}\left[\nabla_{\boldsymbol{\theta}} f_0(\boldsymbol{\theta}; \boldsymbol{\phi}(s, a))^{\mathsf{T}} \nabla_{\boldsymbol{\theta}} f_0(\boldsymbol{\theta}; \boldsymbol{\phi}(s', \tilde{a}))\right]
$$

$$
- \|\hat{\boldsymbol{\theta}}\|^2 \mathbb{E}\left[\nabla_{\boldsymbol{\theta}} f_0(\boldsymbol{\theta}; \boldsymbol{\phi}(s, a))^{\mathsf{T}}\left[\frac{1}{2S} \sum_{\tilde{s} \in \mathcal{S}} \nabla_{\boldsymbol{\theta}} f_0(\boldsymbol{\theta}; \boldsymbol{\phi}(\tilde{s}, 0)) + f_0(\boldsymbol{\theta}; \boldsymbol{\phi}(\tilde{s}, 1))\right]\right]
$$

$$
- \|\hat{\boldsymbol{\theta}}\|^2 \mathbb{E}\left[\nabla_{\boldsymbol{\theta}} f_0(\boldsymbol{\theta}; \boldsymbol{\phi}(s, a))^{\mathsf{T}} \nabla_{\boldsymbol{\theta}} f_0(\boldsymbol{\theta}; \boldsymbol{\phi}(s, a))\right]
$$

where $(b_1)$ holds since $\nabla_{\boldsymbol{\theta}} f_0(\boldsymbol{\theta}; \boldsymbol{\phi}(s, a)) = \nabla_{\boldsymbol{\theta}} f_0(\boldsymbol{\theta}_0^*; \boldsymbol{\phi}(s, a))$ as in (26), $(b_2)$ is due to the fact that $\max_{a_1} f_0(\boldsymbol{\theta}; \boldsymbol{\phi}(s', a_1)) - \max_{a_1} f_0(\boldsymbol{\theta}_0^*; \boldsymbol{\phi}(s', a_2)) \leq \max_{a'}\left[f_0(\boldsymbol{\theta}; \boldsymbol{\phi}(s', a')) - f_0(\boldsymbol{\theta}_0^*; \boldsymbol{\phi}(s', a'))\right]$, and $(b_3)$ holds due to the fact that $\hat{\boldsymbol{\theta}}^{\mathsf{T}} \mathbb{E}\left[\nabla_{\boldsymbol{\theta}} f_0(\boldsymbol{\theta}; \boldsymbol{\phi}(s, a))[(1-a)(\lambda - y_0(\boldsymbol{\theta}_0^*))]\right] \leq 0$ since a larger Whittle index $\lambda$ will choose the action $a = 1$. Notice that the $\tilde{a}$ in $(b_4)$ represents the action $a'$ which maximizes $f_0(\boldsymbol{\theta}; \boldsymbol{\phi}(s', a')) - f_0(\boldsymbol{\theta}_0^*; \boldsymbol{\phi}(s', a'))$. Due to the definition of $\nabla_{\boldsymbol{\theta}} f_0(\boldsymbol{\theta}; \boldsymbol{\phi}(s, a))$ in (26), we show that $\mathbb{E}[\hat{\boldsymbol{\theta}}^{\mathsf{T}} h_0(X, \boldsymbol{\theta}, \lambda)] \leq 0$.

2) Next, we show that there exists a constant $\mu_2 > 0$ such that $\mathbb{E}[\hat{\lambda} g_0(X, \boldsymbol{\theta}, \lambda)] \leq -\mu_2 \|\hat{\lambda}\|^2$. According to the definition of $g_0(\boldsymbol{\theta})$, i.e., $g_0(\boldsymbol{\theta}) := f_0(\boldsymbol{\theta}; \boldsymbol{\phi}(s, 1)) - f_0(\boldsymbol{\theta}; \boldsymbol{\phi}(s, 0))$. Since $y_0(\boldsymbol{\theta})$ is the solution of $\lambda$ such that $f_0(\boldsymbol{\theta}; \boldsymbol{\phi}(s, 1)) = f_0(\boldsymbol{\theta}; \boldsymbol{\phi}(s, 0))$, the signs of $\hat{\lambda} := \lambda - y_0(\theta)$ and $f_0(\boldsymbol{\theta}; \boldsymbol{\phi}(s, 1)) - f_0(\boldsymbol{\theta}; \boldsymbol{\phi}(s, 0))$ are always opposite. Hence, we have $\mathbb{E}[\hat{\lambda} g_0(X, \boldsymbol{\theta}, \lambda)] \leq 0$, which completes the proof.

$\square$

## C.5 Proof of Lemma 5

*Proof.* Under Lemma 1, we have

$$\|h_0(X, \boldsymbol{\theta}, \lambda) - h_0(X, \boldsymbol{\theta}^*, \lambda^*)\| \le 3\|\boldsymbol{\theta} - \boldsymbol{\theta}^*\| + \|\boldsymbol{\lambda} - \boldsymbol{\lambda}^*\|. \tag{29}$$

Let $L = \max(3, \max_X h_0(X, \boldsymbol{\theta}^*, \lambda^*))$, then according to (29), we have

$$\|h_0(X, \boldsymbol{\theta}, \lambda)\| \le L(\|\boldsymbol{\theta} - \boldsymbol{\theta}^*\| + \|\boldsymbol{\lambda} - \boldsymbol{\lambda}^*\| + 1).$$

Denote $h_0^i(X, \boldsymbol{\theta}, \lambda)$ as the $i$-th element of $h_0(X, \boldsymbol{\theta}, \lambda)$. Following [15], we can show that $\boldsymbol{\theta} \in \mathbb{R}^{md}$, $\lambda \in \mathbb{R}^1$, and $x \in \mathcal{X}$,

$$\|\mathbb{E}[h_0(X_k, \boldsymbol{\theta}, \lambda)|X_0 = x] - \mathbb{E}_\mu[h_0(X, \boldsymbol{\theta}, \lambda)]\|$$

$$\le \sum_{i=1}^{md} |\mathbb{E}[h_i(X_k, \boldsymbol{\theta}, \lambda)|X_0 = x] - \mathbb{E}_\mu[h_0^i(X, \boldsymbol{\theta}, \lambda)]|$$

$$\le 2L(\|\boldsymbol{\theta} - \boldsymbol{\theta}^*\| + \|\lambda - \lambda^*\| + 1) \sum_{i=1}^{md} \left| \mathbb{E}\left[ \frac{h_0^i(X_k, \boldsymbol{\theta}, \lambda)}{2L(\|\boldsymbol{\theta} - \boldsymbol{\theta}^*\| + \|\lambda - \lambda^*\| + 1)} \middle| X_0 = x \right] \right.$$

$$\left. - \mathbb{E}_\mu\left[ \frac{h_0^i(X, \boldsymbol{\theta}, \lambda)}{2L(\|\boldsymbol{\theta} - \boldsymbol{\theta}^*\| + \|\lambda - \lambda^*\| + 1)} \right] \right|$$

$$\le 2L(\|\boldsymbol{\theta} - \boldsymbol{\theta}^*\| + \|\lambda - \lambda^*\| + 1)mdC\rho^k,$$

where the last inequality holds due to Assumption 1. To guarantee $2L(\|\boldsymbol{\theta} - \boldsymbol{\theta}^*\| + \|\lambda - \lambda^*\| + 1)mdC\rho^k \le \delta(\|\boldsymbol{\theta} - \boldsymbol{\theta}^*\| + \|\lambda - \lambda^*\| + 1)$, we have

$$\tau_\delta \le \frac{\log(1/\delta) + \log(2LCmd)}{\log(1/\rho)},$$

which completes the proof.

$\square$

## C.6 Proof of Lemma 6

*Proof.* Based on the definition of $M(\boldsymbol{\theta}_k, \lambda_k)$ in (15), we have

$$M(\boldsymbol{\theta}_k, \lambda_k) := \frac{\eta_k}{\alpha_k}\|\boldsymbol{\theta}_k - \boldsymbol{\theta}^*\|^2 + \|\lambda_k - y(\boldsymbol{\theta}_k)\|^2$$

$$= \frac{\eta_k}{\alpha_k}\|\boldsymbol{\theta}_k - \boldsymbol{\theta}_0^* + \boldsymbol{\theta}_0^* - \boldsymbol{\theta}^*\|^2 + \|\lambda_k - y_0(\boldsymbol{\theta}_k) + y_0(\boldsymbol{\theta}_k) - y(\boldsymbol{\theta}_k)\|^2$$

$$\le \frac{2\eta_k}{\alpha_k}(\|\boldsymbol{\theta}_k - \boldsymbol{\theta}_0^*\|^2 + \|\boldsymbol{\theta}_0^* - \boldsymbol{\theta}^*\|^2) + 2(\|\lambda_k - y_0(\boldsymbol{\theta}_k)\|^2 + \|y_0(\boldsymbol{\theta}_k) - y(\boldsymbol{\theta}_k)\|^2)$$

$$= 2\hat{M}(\boldsymbol{\theta}_k, \lambda_k) + \frac{2\eta_k}{\alpha_k}\|\boldsymbol{\theta}_0^* - \boldsymbol{\theta}^*\|^2 + 2\|y_0(\boldsymbol{\theta}_k) - y(\boldsymbol{\theta}_k)\|^2$$

$$\le 2\hat{M}(\boldsymbol{\theta}_k, \lambda_k) + \frac{2\eta_k c_0^2}{\alpha_k}\|span(f_0(\boldsymbol{\theta}_0^*) - f(\boldsymbol{\theta}^*))\|^2 + 2\|y_0(\boldsymbol{\theta}_k) - y(\boldsymbol{\theta}_k)\|^2, \tag{30}$$

where the first inequality holds based on $\|\boldsymbol{x} + \boldsymbol{y}\|^2 \le 2\|\boldsymbol{x}\|^2 + 2\|\boldsymbol{y}\|^2$, and the second inequality holds based on Assumption 3. Next, we bound $\|span(f_0(\boldsymbol{\theta}_0^*) - f(\boldsymbol{\theta}^*))\|$ as follows

$$\|span(f_0(\boldsymbol{\theta}_0^*) - f(\boldsymbol{\theta}^*))\| = \|span(f_0(\boldsymbol{\theta}_0^*) - \Pi_\mathcal{F}f(\boldsymbol{\theta}^*) + \Pi_\mathcal{F}f(\boldsymbol{\theta}^*) - f(\boldsymbol{\theta}^*))\|$$

$$\le \|span(f_0(\boldsymbol{\theta}_0^*) - \Pi_\mathcal{F}f(\boldsymbol{\theta}^*))\| + \|span(\Pi_\mathcal{F}f(\boldsymbol{\theta}^*) - f(\boldsymbol{\theta}^*))\|$$

$$= \|span(\Pi_\mathcal{F}\mathcal{T}f_0(\boldsymbol{\theta}_0^*) - \Pi_\mathcal{F}\mathcal{T}f(\boldsymbol{\theta}^*))\| + \|span(\Pi_\mathcal{F}f(\boldsymbol{\theta}^*) - f(\boldsymbol{\theta}^*))\|$$

$$\le \kappa\|span(f_0(\boldsymbol{\theta}_0^*) - f(\boldsymbol{\theta}^*))\| + \|span(\Pi_\mathcal{F}f(\boldsymbol{\theta}^*) - f(\boldsymbol{\theta}^*))\|, \tag{31}$$

where the last inequality follows (24). This indicates that

$$\|span(f_0(\boldsymbol{\theta}_0^*) - f(\boldsymbol{\theta}^*))\|^2 \le \frac{1}{(1-\kappa)^2}\|span(\Pi_\mathcal{F}f(\boldsymbol{\theta}^*) - f(\boldsymbol{\theta}^*))\|^2. \tag{32}$$

We further bound $\|y_0(\boldsymbol{\theta}_k) - y(\boldsymbol{\theta}_k)\|^2$ as follows

$$\|y_0(\boldsymbol{\theta}_k) - y(\boldsymbol{\theta}_k)\|^2 = \Big\| \sum_{s'} p(s'|s,1) \max_{a_1} f_0(\boldsymbol{\theta}_k; \boldsymbol{\phi}(s', a_1)) - \sum_{s'} p(s'|s,0) \max_{a_2} f_0(\boldsymbol{\theta}_k; \boldsymbol{\phi}(s', a_2))$$

$$- \sum_{s'} p(s'|s,1) \max_{a_3} f(\boldsymbol{\theta}_k; \boldsymbol{\phi}(s', a_3)) + \sum_{s'} p(s'|s,0) \max_{a_4} f(\boldsymbol{\theta}_k; \boldsymbol{\phi}(s', a_4)) \Big\|^2$$

$$= \Big\| \sum_{s'} p(s'|s,1)(\max_{a_1} f_0(\boldsymbol{\theta}_k; \boldsymbol{\phi}(s', a_1)) - \max_{a_3} f(\boldsymbol{\theta}_k; \boldsymbol{\phi}(s', a_3)))$$

$$- \sum_{s'} p(s'|s,0)(\max_{a_2} f_0(\boldsymbol{\theta}_k; \boldsymbol{\phi}(s', a_2)) - \max_{a_4} f(\boldsymbol{\theta}_k; \boldsymbol{\phi}(s', a_4))) \Big\|^2$$

$$\leq 2\| \max_{(s,a)} f_0(\boldsymbol{\theta}_k; \boldsymbol{\phi}(s, a)) - f(\boldsymbol{\theta}_k; \boldsymbol{\phi}(s, a))\|^2$$

$$\leq 2\mathcal{O}\Big( \frac{c_1^3(\|\boldsymbol{\theta}_0\| + |\lambda_0| + 1)^3}{m^{1/2}} \Big), \tag{33}$$

where the last inequality is due to Lemma 10. Substituting (32) and (33) back to (30) yields the final results.

$$\square$$

## D  Proof of the Theorem 2

To prove Theorem 2, we need the following three key lemmas about the error terms defined in (19).

**Lemma 7.** *Let* $\{\boldsymbol{\theta}_k, \lambda_k\}$ *be generated by (9). Then under Lemmas 1-4, for any* $k \geq \tau$, *we have*

$$\mathbb{E}\Big[ \big\| \tilde{\boldsymbol{\theta}}_{k+1} \big\|^2 | \mathcal{F}_{k-\tau} \Big] \leq (1 + 150\alpha_k^2 + \eta_k/\alpha_k - 2\alpha_k\mu_1)\mathbb{E}\Big[ \big\| \hat{\boldsymbol{\theta}}_k \big\|^2 | \mathcal{F}_{k-\tau} \Big] + 6\alpha_k^2 \mathbb{E}\Big[ \big\| \hat{\lambda}_k \big\|^2 | \mathcal{F}_{k-\tau} \Big]$$

$$+ \frac{\alpha_k^3}{\eta_k} \mathcal{O}\Big( c_1^3(\|\boldsymbol{\theta}_0\| + |\lambda_0| + 1)^3 \cdot m^{-1/2} \Big). \tag{34}$$

*Proof.* According to (19), we have $\hat{\boldsymbol{\theta}}_{k+1} := \boldsymbol{\theta}_{k+1} - \boldsymbol{\theta}_0^* = \hat{\boldsymbol{\theta}}_k + \alpha_k h(X_k, \boldsymbol{\theta}_k, \lambda_k)$, which leads to

$$\big\| \hat{\boldsymbol{\theta}}_{k+1} \big\|^2 = \big\| \hat{\boldsymbol{\theta}}_k \big\|^2 + 2\alpha_k \hat{\boldsymbol{\theta}}_k^\mathsf{T} h(X_k, \boldsymbol{\theta}_k, \lambda_k) + \big\| \alpha_k h(X_k, \boldsymbol{\theta}_k, \lambda_k) \big\|^2$$

$$= \big\| \hat{\boldsymbol{\theta}}_k \big\|^2 + 2\alpha_k \hat{\boldsymbol{\theta}}_k^\mathsf{T} (h(X_k, \boldsymbol{\theta}_k, \lambda_k) - h_0(X_k, \boldsymbol{\theta}_k, \lambda_k)) + 2\alpha_k \hat{\boldsymbol{\theta}}_k^\mathsf{T} h_0(X_k, \boldsymbol{\theta}_k, \lambda_k)$$

$$+ \alpha_k^2 \|h(X_k, \boldsymbol{\theta}_k, \lambda_k) - h_0(X_k, \boldsymbol{\theta}_k, \lambda_k) + h_0(X_k, \boldsymbol{\theta}_k, \lambda_k)\|^2$$

$$\leq \big\| \hat{\boldsymbol{\theta}}_k \big\|^2 + 2\alpha_k \hat{\boldsymbol{\theta}}_k^\mathsf{T} (h(X_k, \boldsymbol{\theta}_k, \lambda_k) - h_0(X_k, \boldsymbol{\theta}_k, \lambda_k)) + 2\alpha_k \hat{\boldsymbol{\theta}}_k^\mathsf{T} h_0(X_k, \boldsymbol{\theta}_k, \lambda_k)$$

$$+ 2\alpha_k^2 \|h(X_k, \boldsymbol{\theta}_k, \lambda_k) - h_0(X_k, \boldsymbol{\theta}_k, \lambda_k)\|^2 + 2\alpha_k^2 \|h_0(X_k, \boldsymbol{\theta}_k, \lambda_k)\|^2. \tag{35}$$

The above inequality holds due to the fact that $\|\boldsymbol{x} + \boldsymbol{y}\|^2 \leq 2\|\boldsymbol{x}\|^2 + 2\|\boldsymbol{y}\|^2$. Taking expectations of $\|\hat{\boldsymbol{\theta}}_{k+1}\|^2$ w.r.t $\mathcal{F}_{k-\tau}$ yields

$$\mathbb{E}\Big[ \|\hat{\boldsymbol{\theta}}_{k+1}\|^2 | \mathcal{F}_{k-\tau} \Big] \leq \mathbb{E}\Big[ \big\| \hat{\boldsymbol{\theta}}_k \big\|^2 | \mathcal{F}_{k-\tau} \Big] + 2\alpha_k \mathbb{E}\Big[ \hat{\boldsymbol{\theta}}_k^\mathsf{T} h_0(X_k, \boldsymbol{\theta}_k, \lambda_k) | \mathcal{F}_{k-\tau} \Big]$$

$$+ \underbrace{2\alpha_k^2 \mathbb{E}\Big[ \big\| h_0(X_k, \boldsymbol{\theta}_k, \lambda_k) \big\|^2 | \mathcal{F}_{k-\tau} \Big]}_{\text{Term}_1}$$

$$+ \underbrace{2\alpha_k \mathbb{E}\Big[ \hat{\boldsymbol{\theta}}_k^\mathsf{T} (h(X_k, \boldsymbol{\theta}_k, \lambda_k) - h_0(X_k, \boldsymbol{\theta}_k, \lambda_k)) | \mathcal{F}_{k-\tau} \Big]}_{\text{Term}_2}$$

$$+ \underbrace{2\alpha_k^2 \mathbb{E}\Big[ \big\| h(X_k, \boldsymbol{\theta}_k, \lambda_k) - h_0(X_k, \boldsymbol{\theta}_k, \lambda_k) \big\|^2 | \mathcal{F}_{k-\tau} \Big]}_{\text{Term}_3}$$

$$\leq \mathbb{E}\left[\left\|\hat{\boldsymbol{\theta}}_k\right\|^2 |\mathcal{F}_{k-\tau}\right] - 2\alpha_k\mu_1\mathbb{E}\left[\left\|\tilde{\boldsymbol{\theta}}_k\right\|^2 |\mathcal{F}_{k-\tau}\right] + \text{Term}_1 + \text{Term}_2 + \text{Term}_3, \quad (36)$$

where the last inequality is due to Lemma 4. Next, we bound each individual term. $\text{Term}_1$ is bounded as

$$
\begin{aligned}
\text{Term}_1 &= 2\alpha_k^2 \mathbb{E}\left[\left\|h_0(X_k, \boldsymbol{\theta}_k, \lambda_k)\right\|^2 |\mathcal{F}_{k-\tau}\right] \\
&\stackrel{(c_1)}{=} 2\alpha_k^2 \mathbb{E}\Big[\big\|h_0(X_k, \boldsymbol{\theta}_k, \lambda_k) - h_0(X_k, \boldsymbol{\theta}_k, y_0(\boldsymbol{\theta}_k)) + h_0(X_k, \boldsymbol{\theta}_k, y_0(\boldsymbol{\theta}_k)) \\
&\qquad - h_0(X_k, \boldsymbol{\theta}_0^*, y_0(\boldsymbol{\theta}_0^*)) + h_0(X_k, \boldsymbol{\theta}_0^*, y_0(\boldsymbol{\theta}_0^*)) - H_0(\boldsymbol{\theta}_0^*, y_0(\boldsymbol{\theta}_0^*))\big\|^2 |\mathcal{F}_{k-\tau}\Big] \\
&\stackrel{(c_2)}{\leq} 6\alpha_k^2 \mathbb{E}\left[\left\|h_0(X_k, \boldsymbol{\theta}_k, \lambda_k) - h_0(X_k, \boldsymbol{\theta}_k, y_0(\boldsymbol{\theta}_k))\right\|^2 |\mathcal{F}_{k-\tau}\right] \\
&\qquad + 6\alpha_k^2 \mathbb{E}\left[\left\|h_0(X_k, \boldsymbol{\theta}_k, y_0(\boldsymbol{\theta}_k)) - h_0(X_k, \boldsymbol{\theta}_0^*, y_0(\boldsymbol{\theta}_0^*))\right\|^2 |\mathcal{F}_{k-\tau}\right] \\
&\qquad + 6\alpha_k^2 \mathbb{E}\left[\left\|h_0(X_k, \boldsymbol{\theta}_0^*, y_0(\boldsymbol{\theta}_0^*)) - H_0(\boldsymbol{\theta}_0^*, y_0(\boldsymbol{\theta}_0^*))\right\|^2 |\mathcal{F}_{k-\tau}\right] \\
&\stackrel{(c_3)}{\leq} 6\alpha_k^2 \mathbb{E}\left[\left\|\hat{\lambda}_k\right\|^2 |\mathcal{F}_{k-\tau}\right] + 150\alpha_k^2 \mathbb{E}\left[\left\|\hat{\boldsymbol{\theta}}_k\right\|^2 |\mathcal{F}_{k-\tau}\right], \quad (37)
\end{aligned}
$$

where $(c_1)$ holds due to $H_0(\boldsymbol{\theta}_0, y_0(\boldsymbol{\theta}_0^*)) = 0$, $(c_2)$ follows from the triangular inequality, and $(c_3)$ follows from the Lipschitz continuity of $h_0(X, \boldsymbol{\theta}, \lambda)$ in Lemma 1.

$\text{Term}_2$ is bounded as

$$
\begin{aligned}
\text{Term}_2 &= 2\alpha_k \mathbb{E}\left[\hat{\boldsymbol{\theta}}_k^\mathsf{T}(h(X_k, \boldsymbol{\theta}_k, \lambda_k) - h_0(X_k, \boldsymbol{\theta}_k, \lambda_k)) |\mathcal{F}_{k-\tau}\right] \\
&\stackrel{(c_4)}{\leq} \frac{\eta_k}{\alpha_k} \mathbb{E}\left[\left\|\hat{\boldsymbol{\theta}}_k\right\|^2 |\mathcal{F}_{k-\tau}\right] + \frac{\alpha_k^3}{\eta_k} \mathbb{E}\left[\left\|h(X_k, \boldsymbol{\theta}_k, \lambda_k) - h_0(X_k, \boldsymbol{\theta}_k, \lambda_k)\right\|^2 |\mathcal{F}_{k-\tau}\right] \\
&\stackrel{(c_5)}{\leq} \frac{\eta_k}{\alpha_k} \mathbb{E}\left[\left\|\hat{\boldsymbol{\theta}}_k\right\|^2 |\mathcal{F}_{k-\tau}\right] + \frac{\alpha_k^2}{\eta_k} \mathcal{O}\left(c_1^3(\|\boldsymbol{\theta}_0\| + |\lambda_0| + 1)^3 \cdot m^{-1/2}\right), \quad (38)
\end{aligned}
$$

where $(c_4)$ holds due to the fact that $2\boldsymbol{x}^\mathsf{T}\boldsymbol{y} \leq \|\boldsymbol{x}\|^2 + \|\boldsymbol{y}\|^2$ and $(c_5)$ is due to Lemma 10.

$\text{Term}_3$ is bounded as

$$
\begin{aligned}
\text{Term}_3 &= 2\alpha_k^2 \mathbb{E}\left[\left\|h(X_k, \boldsymbol{\theta}_k, \lambda_k) - h_0(X_k, \boldsymbol{\theta}_k, \lambda_k)\right\|^2 |\mathcal{F}_{k-\tau}\right] \\
&\stackrel{(c_6)}{\leq} 2\alpha_k^2 \mathcal{O}\left(c_1^3(\|\boldsymbol{\theta}_0\| + |\lambda_0| + 1)^3 \cdot m^{-1/2}\right), \quad (39)
\end{aligned}
$$

where $(c_6)$ comes from Lemma 10. Substituting $\text{Term}_1$, $\text{Term}_2$, and $\text{Term}_3$ back into (36) leads to the desired result in (34), which is

$$
\begin{aligned}
\mathbb{E}\left[\|\hat{\boldsymbol{\theta}}_{k+1}\|^2 |\mathcal{F}_{k-\tau}\right] &\leq \mathbb{E}\left[\left\|\hat{\boldsymbol{\theta}}_k\right\|^2 |\mathcal{F}_{k-\tau}\right] - 2\alpha_k\mu_1 \mathbb{E}\left[\left\|\tilde{\boldsymbol{\theta}}_k\right\|^2 |\mathcal{F}_{k-\tau}\right] \\
&\quad + 6\alpha_k^2 \mathbb{E}\left[\left\|\hat{\lambda}_k\right\|^2 |\mathcal{F}_{k-\tau}\right] + 150\alpha_k^2 \mathbb{E}\left[\left\|\hat{\boldsymbol{\theta}}_k\right\|^2 |\mathcal{F}_{k-\tau}\right] \\
&\quad + \frac{\eta_k}{\alpha_k} \mathbb{E}\left[\left\|\hat{\boldsymbol{\theta}}_k\right\|^2 |\mathcal{F}_{k-\tau}\right] + \frac{\alpha_k^3}{\eta_k} \mathcal{O}\left(c_1^3(\|\boldsymbol{\theta}_0\| + |\lambda_0| + 1)^3 \cdot m^{-1/2}\right) \\
&\quad + 2\alpha_k^2 \mathcal{O}\left(c_1^3(\|\boldsymbol{\theta}_0\| + |\lambda_0| + 1)^3 \cdot m^{-1/2}\right) \\
&= (1 + 150\alpha_k^2 + \eta_k/\alpha_k - 2\alpha_k\mu_1)\mathbb{E}\left[\left\|\hat{\boldsymbol{\theta}}_k\right\|^2 |\mathcal{F}_{k-\tau}\right] + 6\alpha_k^2 \mathbb{E}\left[\left\|\hat{\lambda}_k\right\|^2 |\mathcal{F}_{k-\tau}\right] \\
&\quad + (\alpha_k^3/\eta_k + 2\alpha_k^2)\mathcal{O}\left(c_1^3(\|\boldsymbol{\theta}_0\| + |\lambda_0| + 1)^3 \cdot m^{-1/2}\right).
\end{aligned}
$$

By neglecting higher order infinitesimal, we have the inequality in (34). This completes the proof. $\quad\square$

**Lemma 8.** *Let $\{\boldsymbol{\theta}_k, \lambda_k\}$ be generated by (9). Then under Lemmas 1-4, for any $k \geq \tau$, we have*

$$\mathbb{E}\left[\left\|\hat{\lambda}_{k+1}\right\|^2 \Big| \mathcal{F}_{k-\tau}\right] \leq (1 - 2\eta_k\mu_2 + \alpha_k\eta_k + 24\alpha_k^2 + \frac{\eta_k}{\alpha_k} - \frac{2\eta_k^2\mu_k}{\alpha_k} + \eta_k^2 + \frac{24\alpha_k^3}{\eta_k})\mathbb{E}\left[\left\|\hat{\lambda}_k\right\|^2 \Big| \mathcal{F}_{k-\tau}\right]$$

$$+ (600\alpha_k^2 + 8\eta_k^2 + \frac{8\eta_k^3}{\alpha_k} + \frac{600\alpha_k^3}{\eta_k})\mathbb{E}\left[\left\|\hat{\boldsymbol{\theta}}_k\right\|^2 \Big| \mathcal{F}_{k-\tau}\right]$$

$$+ \frac{\eta_k}{\alpha_k}\mathcal{O}\left(c_1^3(\|\boldsymbol{\theta}_0\| + |\lambda_0| + 1)^3 \cdot m^{-1/2}\right). \tag{40}$$

*Proof.* According to the definition in (14), we have

$$\hat{\lambda}_{k+1} = \lambda_{k+1} - y_0(\boldsymbol{\theta}_{k+1})$$
$$= \hat{\lambda}_k + \eta_k g(\boldsymbol{\theta}_k) + y_0(\boldsymbol{\theta}_k) - y_0(\boldsymbol{\theta}_{k+1}),$$

which leads to

$$\left\|\hat{\lambda}_{k+1}\right\|^2 = \left\|\hat{\lambda}_k + \eta_k g(\boldsymbol{\theta}_k) + y_0(\boldsymbol{\theta}_k) - y_0(\boldsymbol{\theta}_{k+1})\right\|^2$$

$$= \underbrace{\left\|\hat{\lambda}_k + \eta_k g(\boldsymbol{\theta}_k)\right\|^2}_{\text{Term}_1} + \underbrace{\left\|y_0(\boldsymbol{\theta}_k) - y_0(\boldsymbol{\theta}_{k+1})\right\|^2}_{\text{Term}_2}$$

$$+ \underbrace{2\left(\hat{\lambda}_k + \eta_k g(\boldsymbol{\theta}_k)\right)\left(y_0(\boldsymbol{\theta}_k) - y_0(\boldsymbol{\theta}_{k+1})\right)}_{\text{Term}_3}. \tag{41}$$

The second equality is due to the fact that $\|\boldsymbol{x} + \boldsymbol{y}\|^2 = \|\boldsymbol{x}\|^2 + \|\boldsymbol{y}\|^2 + 2\boldsymbol{x}^\mathsf{T}\boldsymbol{y}$. We next analyze the conditional expectation of each term in $\left\|\hat{\lambda}_{k+1}\right\|^2$ on $\mathcal{F}_{k-\tau}$. We first focus on Term$_1$.

$$\mathbb{E}\left[\text{Term}_1 | \mathcal{F}_{k-\tau}\right]$$

$$= \mathbb{E}\left[\left\|\hat{\lambda}_k\right\|^2 + 2\eta_k\hat{\lambda}_k g(\boldsymbol{\theta}_k) + \left\|\eta_k g(\boldsymbol{\theta}_k)\right\|^2 \Big| \mathcal{F}_{k-\tau}\right]$$

$$= \mathbb{E}\left[\left\|\hat{\lambda}_k\right\|^2 + 2\eta_k\hat{\lambda}_k g_0(\boldsymbol{\theta}_k) + 2\eta_k\hat{\lambda}_k(g(\boldsymbol{\theta}_k) - g_0(\boldsymbol{\theta}_k)) + \eta_k^2\left\|g(\boldsymbol{\theta}_k) - g_0(\boldsymbol{\theta}_k) + g_0(\boldsymbol{\theta}_k)\right\|^2 \Big| \mathcal{F}_{k-\tau}\right]$$

$$\leq \mathbb{E}\left[\left\|\hat{\lambda}_k\right\|^2 \Big| \mathcal{F}_{k-\tau}\right] + 2\eta_k\mathbb{E}\left[\hat{\lambda}_k g_0(\boldsymbol{\theta}_k)|\mathcal{F}_{k-\tau}\right] + 2\eta_k\mathbb{E}\left[\hat{\lambda}_k(g(\boldsymbol{\theta}_k) - g_0(\boldsymbol{\theta}_k))|\mathcal{F}_{k-\tau}\right]$$

$$+ 2\eta_k^2\mathbb{E}\left[\left\|g(\boldsymbol{\theta}_k) - g_0(\boldsymbol{\theta}_k)\right\|^2 \Big| \mathcal{F}_{k-\tau}\right] + 2\eta_k^2\mathbb{E}\left[\left\|g_0(\boldsymbol{\theta}_k)\right\|^2 \Big| \mathcal{F}_{k-\tau}\right]$$

$$\stackrel{(d_1)}{=} \mathbb{E}\left[\left\|\hat{\lambda}_k\right\|^2 \Big| \mathcal{F}_{k-\tau}\right] + 2\eta_k\mathbb{E}\left[\hat{\lambda}_k g_0(\boldsymbol{\theta}_k)|\mathcal{F}_{k-\tau}\right] + 2\eta_k\mathbb{E}\left[\hat{\lambda}_k(g(\boldsymbol{\theta}_k) - g_0(\boldsymbol{\theta}_k))|\mathcal{F}_{k-\tau}\right]$$

$$+ 2\eta_k^2\mathbb{E}\left[\left\|g(\boldsymbol{\theta}_k) - g_0(\boldsymbol{\theta}_k)\right\|^2 \Big| \mathcal{F}_{k-\tau}\right] + 2\eta_k^2\mathbb{E}\left[\left\|g_0(\boldsymbol{\theta}_k) - g_0(\boldsymbol{\theta}_0^*)\right\|^2 \Big| \mathcal{F}_{k-\tau}\right]$$

$$\stackrel{(d_2)}{=} \mathbb{E}\left[\left\|\hat{\lambda}_k\right\|^2 \Big| \mathcal{F}_{k-\tau}\right] - 2\eta_k\mu_2\mathbb{E}\left[\left\|\hat{\lambda}_k\right\|^2 \Big| \mathcal{F}_{k-\tau}\right] + 8\eta_k^2\mathbb{E}\left[\left\|\hat{\boldsymbol{\theta}}_k\right\|^2 \Big| \mathcal{F}_{k-\tau}\right]$$

$$+ 2\eta_k\mathbb{E}\left[\hat{\lambda}_k(g(\boldsymbol{\theta}_k) - g_0(\boldsymbol{\theta}_k))|\mathcal{F}_{k-\tau}\right] + 2\eta_k^2\mathbb{E}\left[\left\|g(\boldsymbol{\theta}_k) - g_0(\boldsymbol{\theta}_k)\right\|^2 \Big| \mathcal{F}_{k-\tau}\right]$$

$$\stackrel{(d_3)}{\leq} \mathbb{E}\left[\left\|\hat{\lambda}_k\right\|^2 \Big| \mathcal{F}_{k-\tau}\right] - 2\eta_k\mu_2\mathbb{E}\left[\left\|\hat{\lambda}_k\right\|^2 \Big| \mathcal{F}_{k-\tau}\right] + 8\eta_k^2\mathbb{E}\left[\left\|\hat{\boldsymbol{\theta}}_k\right\|^2 \Big| \mathcal{F}_{k-\tau}\right]$$

$$+ \alpha_k\eta_k\mathbb{E}\left[\left\|\hat{\lambda}_k\right\|^2 \Big| \mathcal{F}_{k-\tau}\right] + (4\eta_k/\alpha_k + 8\eta_k^2)\mathcal{O}\left(c_1^3(\|\boldsymbol{\theta}_0\| + |\lambda_0| + 1)^3 \cdot m^{-1/2}\right),$$

where $(d_1)$ follows from $g_0(\boldsymbol{\theta}_0^*) = 0$, $(d_2)$ holds due to Lemma 4 and the Lipschitz continuity of $y_0$ in Lemma 3, and $(d_3)$ comes from Lemma 10. For Term$_2$, we have

$$\mathbb{E}\left[\text{Term}_2 | \mathcal{F}_{k-\tau}\right] = \mathbb{E}\left[\left\|y_0(\boldsymbol{\theta}_k) - y_0(\boldsymbol{\theta}_{k+1})\right\|^2 \Big| \mathcal{F}_{k-\tau}\right]$$

$$= 4\mathbb{E}\left[\left\|\boldsymbol{\theta}_k - \boldsymbol{\theta}_{k+1}\right\|^2 |\mathcal{F}_{k-\tau}\right]$$

$$= 4\alpha_k^2 \mathbb{E}\left[\left\|h(X_k, \boldsymbol{\theta}_k, \lambda_k)\right\|^2 |\mathcal{F}_{k-\tau}\right]$$

$$= 4\alpha_k^2 \mathbb{E}\left[\left\|h(X_k, \boldsymbol{\theta}_k, \lambda_k) - h_0(X_k, \boldsymbol{\theta}_k, \lambda_k) + h_0(X_k, \boldsymbol{\theta}_k, \lambda_k)\right\|^2 |\mathcal{F}_{k-\tau}\right]$$

$$= 8\alpha_k^2 \mathbb{E}\left[\left\|h_0(X_k, \boldsymbol{\theta}_k, \lambda_k)\right\|^2 |\mathcal{F}_{k-\tau}\right] + 8\alpha_k^2 \mathbb{E}\left[\left\|h(X_k, \boldsymbol{\theta}_k, \lambda_k) - h_0(X_k, \boldsymbol{\theta}_k, \lambda_k)\right\|^2 |\mathcal{F}_{k-\tau}\right]$$

$$\overset{(d_4)}{=} 8\alpha_k^2 \mathbb{E}\Big[\Big\|h_0(X_k, \boldsymbol{\theta}_k, \lambda_k) - h_0(X_k, \boldsymbol{\theta}_k, y_0(\boldsymbol{\theta}_k)) + h_0(X_k, \boldsymbol{\theta}_k, y_0(\boldsymbol{\theta}_k))$$

$$- h_0(X_k, \boldsymbol{\theta}_0^*, y_0(\boldsymbol{\theta}_0^*)) + h_0(X_k, \boldsymbol{\theta}_0^*, y_0(\boldsymbol{\theta}_0^*)) - H_0(\boldsymbol{\theta}_0^*, y_0(\boldsymbol{\theta}_0^*))\Big\|^2 |\mathcal{F}_{k-\tau}\Big]$$

$$+ 8\alpha_k^2 \mathbb{E}\left[\left\|h(X_k, \boldsymbol{\theta}_k, \lambda_k) - h_0(X_k, \boldsymbol{\theta}_k, \lambda_k)\right\|^2 |\mathcal{F}_{k-\tau}\right]$$

$$\overset{(d_5)}{\leq} 24\alpha_k^2 \mathbb{E}\left[\left\|h_0(X_k, \boldsymbol{\theta}_k, \lambda_k) - h_0(X_k, \boldsymbol{\theta}_k, y_0(\boldsymbol{\theta}_k))\right\|^2 |\mathcal{F}_{k-\tau}\right]$$

$$+ 24\alpha_k^2 \mathbb{E}\left[\left\|h_0(X_k, \boldsymbol{\theta}_k, y_0(\boldsymbol{\theta}_k)) - h_0(X_k, \boldsymbol{\theta}_0^*, y_0(\boldsymbol{\theta}_0^*))\right\|^2 |\mathcal{F}_{k-\tau}\right]$$

$$+ 24\alpha_k^2 \mathbb{E}\left[\left\|h_0(X_k, \boldsymbol{\theta}_0^*, y_0(\boldsymbol{\theta}_0^*)) - H_0(\boldsymbol{\theta}_0^*, y_0(\boldsymbol{\theta}_0^*))\right\|^2 |\mathcal{F}_{k-\tau}\right]$$

$$+ 8\alpha_k^2 \mathbb{E}\left[\left\|h(X_k, \boldsymbol{\theta}_k, \lambda_k) - h_0(X_k, \boldsymbol{\theta}_k, \lambda_k)\right\|^2 |\mathcal{F}_{k-\tau}\right]$$

$$\overset{(d_6)}{\leq} 24\alpha_k^2 \mathbb{E}\left[\left\|\hat{\lambda}_k\right\|^2 |\mathcal{F}_{k-\tau}\right] + 600\alpha_k^2 \mathbb{E}\left[\left\|\hat{\boldsymbol{\theta}}_k\right\|^2 |\mathcal{F}_{k-\tau}\right]$$

$$+ 8\alpha_k^2 \mathbb{E}\left[\left\|h(X_k, \boldsymbol{\theta}_k, \lambda_k) - h_0(X_k, \boldsymbol{\theta}_k, \lambda_k)\right\|^2 |\mathcal{F}_{k-\tau}\right]$$

$$\overset{(d_7)}{\leq} 24\alpha_k^2 \mathbb{E}\left[\left\|\hat{\lambda}_k\right\|^2 |\mathcal{F}_{k-\tau}\right] + 600\alpha_k^2 \mathbb{E}\left[\left\|\hat{\boldsymbol{\theta}}_k\right\|^2 |\mathcal{F}_{k-\tau}\right]$$

$$+ 8\alpha_k^2 \mathcal{O}\Big(c_1^3(\|\boldsymbol{\theta}_0\| + |\lambda_0| + 1)^3 \cdot m^{-1/2}\Big) \tag{42}$$

where $(d_4)$ is due to the fact that $H_0(\boldsymbol{\theta}_0^*, y_0(\boldsymbol{\theta}_0^*)) = 0$, $(d_5)$ holds according to $\|\boldsymbol{x} + \boldsymbol{y} + \boldsymbol{z}\|^2 \leq 3\|\boldsymbol{x}\|^2 + 3\|\boldsymbol{y}\|^2 + 3\|\boldsymbol{z}\|^2$ since $g(X_k, f(\boldsymbol{\lambda}^*), \boldsymbol{\lambda}^*) = \mathbf{0}$, $(d_6)$ holds because of the Lipschitz continuity of $h_0$ and $y_0$ in Lemma 1 and Lemma 3, and $(d_7)$ comes from Lemma 10. Next, we have the conditional expectation of Term$_3$ as

$$\mathbb{E}\left[\text{Term}_3 | \mathcal{F}_{k-\tau}\right] = 2\mathbb{E}\left[\left\|\hat{\lambda}_k + \eta_k g(\boldsymbol{\theta}_k)\right\| \cdot \left\|y_0(\boldsymbol{\theta}_k) - y_0(\boldsymbol{\theta}_{k+1})\right\| |\mathcal{F}_{k-\tau}\right]$$

$$\overset{(d_8)}{\leq} \frac{\eta_k}{\alpha_k}\text{Term}_1 + \frac{\alpha_k}{\eta_k}\text{Term}_2$$

$$= \frac{\eta_k}{\alpha_k}\mathbb{E}\left[\left\|\hat{\lambda}_k\right\|^2 |\mathcal{F}_{k-\tau}\right] - \frac{2\eta_k^2 \mu_2}{\alpha_k}\mathbb{E}\left[\left\|\hat{\lambda}_k\right\|^2 |\mathcal{F}_{k-\tau}\right] + \frac{8\eta_k^3}{\alpha_k}\mathbb{E}\left[\left\|\hat{\boldsymbol{\theta}}_k\right\|^2 |\mathcal{F}_{k-\tau}\right]$$

$$+ \eta_k^2 \mathbb{E}\left[\left\|\hat{\lambda}_k\right\|^2 |\mathcal{F}_{k-\tau}\right] + \frac{\eta_k}{\alpha_k}(4\eta_k/\alpha_k + 8\eta_k^2)\mathcal{O}\Big(c_1^3(\|\boldsymbol{\theta}_0\| + |\lambda_0| + 1)^3 \cdot m^{-1/2}\Big)$$

$$+ \frac{24\alpha_k^3}{\eta_k}\mathbb{E}\left[\left\|\hat{\lambda}_k\right\|^2 |\mathcal{F}_{k-\tau}\right] + \frac{600\alpha_k^3}{\eta_k}\mathbb{E}\left[\left\|\hat{\boldsymbol{\theta}}_k\right\|^2 |\mathcal{F}_{k-\tau}\right]$$

$$+ \frac{8\alpha_k^3}{\eta_k}\mathcal{O}\Big(c_1^3(\|\boldsymbol{\theta}_0\| + |\lambda_0| + 1)^3 \cdot m^{-1/2}\Big),$$

where $(d_8)$ holds because $2\mathbf{x}^T\mathbf{y} \leq 1/\beta\|\mathbf{x}\|^2 + \beta\|\mathbf{y}\|^2, \forall \beta > 0$. Summing Term$_1$, Term$_2$, and Term$_3$ and neglecting higher order infinitesimal yield the desired result. $\qquad\square$

Now we are ready to prove the results in Theorem 2. Providing Lemma 8 and Lemma 7, if $\frac{\eta_k}{\alpha_k}$ is non-increasing, we have the following inequality

$$\mathbb{E}\left[\hat{M}(\boldsymbol{\theta}_{k+1}, \lambda_{k+1})\Big|\mathcal{F}_{k-\tau}\right] = \mathbb{E}\left[\frac{\eta_k}{\alpha_k}\left\|\hat{\boldsymbol{\theta}}_{k+1}\right\|^2 + \left\|\hat{\lambda}_{k+1}\right\|^2\Big|\mathcal{F}_{k-\tau}\right]$$

$$\leq \frac{\eta_k}{\alpha_k}(1 - 2\alpha_k\mu_1)\mathbb{E}\left[\left\|\hat{\boldsymbol{\theta}}_k\right\|^2|\mathcal{F}_{k-\tau}\right] + \frac{600\alpha_k^3}{\eta_k}\mathbb{E}\left[\left\|\hat{\boldsymbol{\theta}}_k\right\|^2|\mathcal{F}_{k-\tau}\right]$$

$$+ \frac{8\alpha_k^3}{\eta_k}\mathcal{O}\left(c_1^3(\|\boldsymbol{\theta}_0\| + |\lambda_0| + 1)^3 \cdot m^{-1/2}\right)$$

$$+ (1 - 2\eta_k\mu_2)\mathbb{E}\left[\left\|\hat{\lambda}_k\right\|^2|\mathcal{F}_{k-\tau}\right]$$

$$+ \frac{600\alpha_k^3}{\eta_k}\mathbb{E}\left[\left\|\hat{\lambda}_k\right\|^2|\mathcal{F}_{k-\tau}\right]. \tag{43}$$

Since $(k+1)^2 \cdot \frac{\alpha_k^3}{\eta} = \frac{\alpha_0^3}{\eta_0}(k+1)^{1/3}$, multiplying both sides of (43) with $(k+1)^2$, we have

$$(k+1)^2\mathbb{E}\left[\hat{M}(\boldsymbol{\theta}_{k+1}, \lambda_{k+1})\Big|\mathcal{F}_{k-\tau}\right]$$

$$\leq k^2\mathbb{E}\left[\hat{M}(\boldsymbol{\theta}_k, \lambda_k)\Big|\mathcal{F}_{k-\tau}\right] + \frac{600\alpha_0^3}{\eta_0}(k+1)^{1/3}\left(\left\|\hat{\boldsymbol{\theta}}_k\right\|^2 + \left\|\hat{\lambda}_k\right\|^2\right)$$

$$+ \frac{8\alpha_0^3}{\eta_0}(k+1)^{1/3}\mathcal{O}\left(c_1^3(\|\boldsymbol{\theta}_0\| + |\lambda_0| + 1)^3 \cdot m^{-1/2}\right). \tag{44}$$

Summing (44) from time step $\tau$ to time step $k$, we have

$$(k+1)^2\mathbb{E}\left[\hat{M}(\boldsymbol{\theta}_{k+1}, \lambda_{k+1})\Big|\mathcal{F}_k\right] \leq \tau^2\mathbb{E}\left[\hat{M}(\boldsymbol{\theta}_\tau, \lambda_\tau)\right] + \frac{600\alpha_0^3}{\eta_0}(k+1)^{4/3}\left(\left\|\hat{\boldsymbol{\theta}}_\tau\right\|^2 + \left\|\hat{\lambda}_\tau\right\|^2\right)$$

$$+ \frac{8\alpha_0^3}{\eta_0}(k+1)^{4/3}\mathcal{O}\left(c_1^3(\|\boldsymbol{\theta}_0\| + |\lambda_0| + 1)^3 \cdot m^{-1/2}\right)$$

$$\leq \tau^2\mathbb{E}\left[\hat{M}(\boldsymbol{\theta}_\tau, \lambda_\tau)\right] + \frac{600\alpha_0^3}{\eta_0}\frac{(C_1 + \|\hat{\boldsymbol{\theta}}_0\|)^2 + (2C_1 + \|\hat{\lambda}_0\|)^2}{(k+1)^{-4/3}}$$

$$+ \frac{8\alpha_0^3}{\eta_0}\frac{\mathcal{O}\left(c_1^3(\|\boldsymbol{\theta}_0\| + |\lambda_0| + 1)^3 \cdot m^{-1/2}\right)}{(k+1)^{-4/3}}, \tag{45}$$

where the second inequality holds due to Lemma 9. Finally, dividing both sides by $(k+1)^2$ and moving the constant term into $\mathcal{O}(\cdot)$ yields the results in Theorem 2.

## E  Auxiliary Lemmas

In this part, we present several key lemmas which are needed for the major proofs. We first show the parameters update in (9) is bounded in the following lemma.

**Lemma 9.** *The update of $\boldsymbol{\theta}_k$ and $\lambda_k$ in (9) is bounded with respect to the initial $\boldsymbol{\theta}_0$ and $\lambda_0$, i.e.,*

$$\|\boldsymbol{\theta}_k - \boldsymbol{\theta}_0\| + |\lambda_k - \lambda_0| \leq c_1(\|\boldsymbol{\theta}_0\| + |\lambda_0| + 1),$$

*with $c_1$ be the constant, i.e., $c_1 := \frac{1}{2} + \frac{3}{2}(L_h'\alpha_\tau + L_g'\eta_\tau)(L_h'\alpha_\tau + L_g'\eta_\tau + 1)$.*

*Proof.* Without loss of generality, we assume that

$$L_h' \geq \max(3, \max_{X \in \mathcal{X}}\|h_0(X, 0, 0)\|), \ L_g' \geq \max(2, \max_{X \in \mathcal{X}}\|g_0(X, 0, 0)\|).$$

Then based on triangular inequality and Lemmas 1-2, we have

$$\|h_0(X, \boldsymbol{\theta}, \lambda)\| \leq L_h'(\|\boldsymbol{\theta}\| + |\lambda| + 1), \ \|g_0(X, \boldsymbol{\theta}, \lambda)\| \leq L_g'(\|\boldsymbol{\theta}\| + |\lambda| + 1), \forall \boldsymbol{\theta}, \lambda, X \in \mathcal{X}. \tag{46}$$

Since we have $\boldsymbol{\theta}_{k+1} = \boldsymbol{\theta}_k + \alpha_k h(X_k, \boldsymbol{\theta}_k, \lambda_k)$, we have the following inequality due to Lipschitz continuity of $h$ in (46)

$$\|\boldsymbol{\theta}_{k+1} - \boldsymbol{\theta}_k\| = \alpha_k \|h(X_k, \boldsymbol{\theta}_k, \lambda_k)\| \leq \alpha_k L'_h(\|\boldsymbol{\theta}_k\| + |\lambda_k| + 1). \tag{47}$$

Similarly, we have

$$|\lambda_{k+1} - \lambda_k| = \eta_k |g(X_k, \boldsymbol{\theta}_k, \lambda_k)| \leq \eta_k L'_g(\|\boldsymbol{\theta}_k\| + |\lambda_k| + 1). \tag{48}$$

Due to triangular inequality, adding (47) and (48) leads to

$$\begin{aligned}
\|\boldsymbol{\theta}_{k+1}\| + |\lambda_{k+1}| + 1 &\leq (L'_h \alpha_k + L'_g \eta_k + 1)(\|\boldsymbol{\theta}_k\| + |\lambda_k| + 1) \\
&\leq (L'_h \alpha_0 + L'_g \eta_0 + 1)(\|\boldsymbol{\theta}_k\| + |\lambda_k| + 1),
\end{aligned} \tag{49}$$

where the second inequality holds due to the non-increasing learning rates $\{\alpha_k, \eta_k\}$. Rewriting the above inequality in (49) in a recursive manner yields

$$\|\boldsymbol{\theta}_k\| + \lambda_k + 1 \leq (L'_h \alpha_0 + L'_g \eta_0 + 1)^{k-\tau}(\|\boldsymbol{\theta}_\tau\| + |\lambda_\tau| + 1). \tag{50}$$

Hence, we have

$$\begin{aligned}
\|\boldsymbol{\theta}_k - \boldsymbol{\theta}_{k-\tau}\| + |\lambda_k - \lambda_{k-\tau}| &\leq \sum_{t=k-\tau}^{k-1} \|\boldsymbol{\theta}_{t+1} - \boldsymbol{\theta}_t\| + |\lambda_{t+1} - \lambda_t| \\
&\leq (L'_h \alpha_0 + L'_g \eta_0) \sum_{t=k-\tau}^{k-1} (\|\boldsymbol{\theta}_t\| + |\lambda_t| + 1) \\
&\leq (L'_h \alpha_0 + L'_g \eta_0)(\|\boldsymbol{\theta}_{k-\tau}\| + |\lambda_{k-\tau}| + 1) \sum_{t=k-\tau}^{k-1} (L'_h \alpha_0 + L'_g \eta_0 + 1)^{t-\tau} \\
&= [(L'_h \alpha_0 + L'_g \eta_0 + 1)^\tau - 1](\|\boldsymbol{\theta}_{k-\tau}\| + |\lambda_{k-\tau}| + 1) \\
&\leq (e^{(L'_h \alpha_0 + L'_g \eta_0)\tau} - 1)(\|\boldsymbol{\theta}_{k-\tau}\| + |\lambda_{k-\tau}| + 1) \\
&\leq 2(L'_h \alpha_0 + L'_g \eta_0)\tau(\|\boldsymbol{\theta}_{k-\tau}\| + |\lambda_{k-\tau}| + 1),
\end{aligned}$$

where the last inequality holds when $(L'_h \alpha_0 + L'_g \eta_0)\tau \leq 1/4$. This implies when $k = \tau$, we have

$$\|\boldsymbol{\theta}_\tau - \boldsymbol{\theta}_0\| + |\lambda_\tau - \lambda_0| \leq 2(L'_h \alpha_0 + L'_g \eta_0)\tau(\|\boldsymbol{\theta}_0\| + |\lambda_0| + 1). \tag{51}$$

Similarly, we also have

$$\begin{aligned}
\|\boldsymbol{\theta}_k - \boldsymbol{\theta}_\tau\| + \lambda_k - \lambda_\tau &\leq \sum_{t=\tau}^{k-1} \|\boldsymbol{\theta}_{t+1} - \boldsymbol{\theta}_t\| + \lambda_{t+1} - \lambda_t \\
&\leq \sum_{t=\tau}^{k-1} (L'_h \alpha_t + L'_g \eta_t)(\|\boldsymbol{\theta}_t\| + \lambda_t + 1) \\
&\leq (\|\boldsymbol{\theta}_\tau\| + \lambda_\tau + 1) \sum_{t=\tau}^{k-1} (L'_h \alpha_t + L'_g \eta_t) \prod_{i=0}^{t-\tau} (L'_h \alpha_{\tau+i} + L'_g \eta_{\tau+i} + 1). \tag{52}
\end{aligned}$$

Therefore, the following inequality holds

$$\begin{aligned}
&\|\boldsymbol{\theta}_k - \boldsymbol{\theta}_0\| + |\lambda_k - \lambda_0| \\
&\leq \|\boldsymbol{\theta}_k - \boldsymbol{\theta}_\tau\| + |\lambda_k - \lambda_\tau| + \|\boldsymbol{\theta}_\tau - \boldsymbol{\theta}_0\| + |\lambda_\tau - \lambda_0| \\
&\leq 2(L'_h \alpha_0 + L'_g \eta_0)\tau(\|\boldsymbol{\theta}_0\| + |\lambda_0| + 1) \\
&\quad + (\|\boldsymbol{\theta}_\tau\| + \lambda_\tau + 1) \sum_{t=\tau}^{k-1} (L'_h \alpha_t + L'_g \eta_t) \prod_{i=0}^{t-\tau} (L'_h \alpha_{\tau+i} + L'_g \eta_{\tau+i} + 1) \\
&\leq 2(L'_h \alpha_0 + L'_g \eta_0)\tau(\|\boldsymbol{\theta}_0\| + |\lambda_0| + 1)
\end{aligned}$$

$$+ (2(L_h'\alpha_0 + L_g'\eta_0)\tau + 1) \sum_{t=\tau}^{k-1}(L_h'\alpha_t + L_g'\eta_t) \prod_{i=0}^{t-\tau}(L_h'\alpha_{\tau+i} + L_g'\eta_{\tau+i} + 1)(\|\boldsymbol{\theta}_0\| + |\lambda_0| + 1)$$

$$\leq \left( \frac{1}{2} + \frac{3}{2} \sum_{t=\tau}^{k-1}(L_h'\alpha_t + L_g'\eta_t) \prod_{i=0}^{t-\tau}(L_h'\alpha_{\tau+i} + L_g'\eta_{\tau+i} + 1) \right) (\|\boldsymbol{\theta}_0\| + |\lambda_0| + 1),$$

with the last equality holds when $(L_h'\alpha_0 + L_g'\eta_0)\tau \leq 1/4$. When $\sum_{t=\tau}^{k-1}(L_h'\alpha_t + L_g'\eta_t)\prod_{i=0}^{t-\tau}(L_h'\alpha_{\tau+i} + L_g'\eta_{\tau+i} + 1)$ is non-increasing with $k$, then we can set $c_1$ as $c_1 := \frac{1}{2} + \frac{3}{2}.(L_h'\alpha_\tau + L_g'\eta_\tau)(L_h'\alpha_\tau + L_g'\eta_\tau + 1)$. This completes the proof. $\qquad\square$

Provided Lemma 9, we have the following lemma related with local linearization of Q functions and the original Q functions.

**Lemma 10** (Lemma 5.2 in [13]). *There exists a constant $c_1$ such that*

$$\mathbb{E}\left[ \|h(X_k, \boldsymbol{\theta}_k, \lambda_k) - h_0(X_k, \boldsymbol{\theta}_k, \lambda_k)\|^2 | \mathcal{F}_{k-\tau} \right] \leq \mathcal{O}\left( c_1^3(\|\boldsymbol{\theta}_0\| + |\lambda_0| + 1)^3 \cdot m^{-1/2} \right).$$

Lemma 10 indicates that if the updated parameter is always bounded in a ball with the initialized one as the center and a fixed radius, the local linearized function $f_0(\cdot)$ in (18) and the original neural network approximated function $f(\cdot)$ in (6) have bounded gap, which tends to be zero as the width of hidden layer $m$ grow large. For interested readers, please refer to [13] for detailed proofs of this lemma.

