# OpenReview forum: "Finite-Time Analysis of Whittle Index based Q-Learning for Restless Multi-Armed Bandits with Neural Network Function Approximation"
_NeurIPS.cc/2023/Conference — NeurIPS 2023 poster_

### Official Review · Reviewer_PM6q · 2023-07-02

**Soundness:** 4 excellent
**Presentation:** 2 fair
**Contribution:** 3 good
**Rating:** 7
**Confidence:** 4

**Summary:**

This paper proposes a neural network approach to learning the Whittle index policy. In addition, the paper gives a finite-time analysis for the algorithm, and shows that the algorithm indeed learns the Whittle index values for restless bandits.


**Strengths:**

The paper’s sections are well-written and the technical claims of the paper are sound. In addition, the paper provides sufficient analysis of the two-timescale stochastic approximation used here (to learn the Whittle index values and the action-value function for each arm). Also, the paper offers finite time performance bounds in the neural network setting which is a largely unstudied area.


**Weaknesses:**

While the paper is well-written, I believe it’s better if less remarks/lemmas blocks are used and if they are written as paragraphs. Other than that, the technical details appear correct to the best of my knowledge.


**Questions:**

Question: the neural network is using ReLU activation functions. Would the finite-time analysis be applicable to other activation functions such as tanh?

**Limitations:**

The limitations are addressed by the authors in the paper in regards to the finite-time analysis.

---

> ### Author Rebuttal · Authors · 2023-08-09
>
> Thank you very much for your review and constructive comments, as well as giving the positive rating of our work. Here we would like to address the reviewer's concerns and hope that can help raise the rating of our paper. The detailed responses are as follows:
>
> **Weakness #1:** While the paper is well-written, I believe it’s better if less remarks/lemmas blocks are used and if they are written as paragraphs. Other than that, the technical details appear correct to the best of my knowledge.
>
> **Our Response:**  We thank this reviewer for this suggestion. We will modify those remarks and lemmas in the camera-ready version per your suggestion.
>
> **Question #1:** The neural network is using ReLU activation functions. Would the finite-time analysis be applicable to other activation functions such as tanh?
>
> **Our Response:** We thank this reviewer for this insightful comment. The finite-time analysis of Deep Q-Networks (DQN) [13, 22, 58] and references therein focuses on the ReLU (Rectified Linear Unit) activation function, as it has certain properties that make the analysis tractable. ReLU is piecewise linear and non-saturating, which can simplify the mathematical analysis.
>
> Applying the same analysis to other activation functions like the hyperbolic tangent (tanh) could be more complex. The tanh function is smooth and saturating, meaning that it can squash its input into a small range. This can lead to different behaviors during training, such as vanishing gradients, which might not be captured by an analysis designed for ReLU. It is usually used in classification between two classes and rarely used in deep Q-learning.
>
> To the best of our knowledge, the research for other activation functions on DQN is quite limited, which is an open research problem at current stage.

---

> > ### Comment · Reviewer_PM6q · 2023-08-15
> >
> > I thank the authors for answering my question; I will keep my score at 7 for the paper.

---

> > > ### Author Response · Authors · 2023-08-15
> > > **Thank you!**
> > >
> > > Thank you for your acknowledgement and keeping the positive rating of our paper. Much appreciated!

---

### Official Review · Reviewer_ZyEQ · 2023-07-02

**Soundness:** 3 good
**Presentation:** 3 good
**Contribution:** 3 good
**Rating:** 6
**Confidence:** 4

**Summary:**

This paper presents a neural Q learning method to compute the Whittle indices in restless multi-armed bandit problems. The paper provides an algorithm using two-timescale stochastic approximation (2TSA) to update the parameters in the neural networks and the Whittle indices jointly with different learning rates. The authors also show that the 2TSA method guarantees convergence to the optimal/approximately optimal solution of Whittle indices. One of the major contributions is the breakthrough of not using projection step in the 2TSA algorithm. The algorithm proposed by the authors doesn’t require the functional class to contain the optimal functional approximator. The corresponding approximation guarantee also shows an additional term dependent on the distance of the true optimal to the span of the functional bases used in neural networks. Lastly, the authors provide experiments to show the convergence of the proposed algorithm and empirically verify the assumptions made in the paper.

**Strengths:**

I like the idea of projecting the neural networks using ReLU to linear functions to enable downstream analysis. The paper also generalizes the theoretical analysis in previous work to quantify the impact of approximation error in the functional approximator class. I didn’t go through the appendix, but the proof sketch is clear to me. Overall, the paper is nicely written with new theoretical contribution compared to the previous work. It would be great if the authors can further emphasize and summarize the contributions in the theoretical analysis that are incurred due to not using projection steps. The convergence rate also matches the previous work using projection steps.

**Weaknesses:**

The comparison to the previous algorithm is not clear. I didn’t fully understand why the previous algorithm requires a project step to force the parameters in a bounded set and why your algorithm doesn’t. I believe it is due to the analysis in the convergence guarantee where unbounded parameters can lead to useless bound or divergence. Could you please clarify and emphasis this in the paper more to highlight your contribution more clearly? Especially this seems to be the major contribution of the paper. It deserves a larger portion of the paper to clarify it.

Please also clarify why the definition of Whittle index is different from Whittle et al. and most of the literature. Please see below for more details. I am worried that the different definition can impact the convergence analysis (especially the linearity and Lipschitzness in the proof). Please either justify the use of your definition and any references showing this different definition is valid, or show us that the analysis is not impacted by the definition.

These two are my major concerns of the paper. Please clarify them and I am happy to update my score based on the response.

[Answered by the authors during the rebuttal]

**Questions:**

**Question 1**:
Equation 3: the Whittle index defined by Whittle is defined as the Lagrangian multiplier such that the Q values of action 0 and 1 are identical, i.e., whittle index $\coloneqq  \inf_\lambda \{ Q_n(s,1;\lambda) = Q_n(s,0;\lambda) \}$, where the Q functions are functions of lambda (Lagrangian multiplier) and thus parameterized by lambda. However, your definition in Equation 3 is defined as the difference between Q values, i.e., whittle index $\coloneqq Q_n(s,0) – Q_n(s,1)$, where the Lagrangian multiplier for defining the Q values is also not specified. To my knowledge, this is not equivalent to the original Whittle index definition. The original Whittle index is defined based on the solution to Lagrangian relaxation, but yours doesn’t have this property. Please clarify/justify why you can define Whittle index in this way or provide any references supporting your claim. And please also clarify how this definition impacts you downstream analysis.

I have also checked your references in [3,23] are also aligned with the original Whittle index definition (Whittle index defined such that Q(s,0) = Q(s,1)). Please see Equation (4) in [23] and Equation (11) in [3].

Equation (13) will also be impacted by the definition of Whittle index. So does the analysis in Equation (21) and Lemma 2, which will no longer be linear.


References:

[3] Konstantin E Avrachenkov and Vivek S Borkar. Whittle index based q-learning for restless bandits with average reward. Automatica, 139:110186, 2022.

[23] Jing Fu, Yoni Nazarathy, Sarat Moka, and Peter G Taylor. Towards q-learning the whittle index for restless bandits. In 2019 Australian & New Zealand Control Conference (ANZCC), pages 249–254. IEEE, 2019.

---

**Question 2**:
Do other choices of step size sequence $\alpha$ and $\eta$ work as well?

**Feedback 3**:
Equation 18 is the linearized version of Equation 6 where the coefficient associated to the linear term is defined by the initial point $\theta_0 = [w_{r,0}]$. The definition is correct but slightly unclear.

**Limitations:**

- Different definition of Whittle index: if this is true, then the analysis might be restricted to the definition shown in the paper.
- Linearization: the authors need to linearize the Lyapunov function to enable the theoretical analysis. I believe this is the reason why the authors focus on 2-layer neural networks (only input and output and a ReLU activation function). This can limit the functional approximator class to such neural networks only. For more complex NN structures, the same linearization trick and analysis don't apply anymore.

---

> ### Author Rebuttal · Authors · 2023-08-09
>
> Thank you very much for your review and constructive comments. Here we would like to address the reviewer's concerns and hope that can help raise the rating of our paper. The detailed responses are as follows:
>
> **Weakness \#1:** ... project step...
>
> **Our Response:** Thank you for this insightful comment. The projection step in reinforcement learning originates from [6], which provided the first finite-time performance of TD learning with linear function approximation. Due to the Markovian observation noise, the dependent nature of the data introduces a substantial technical challenge: the algorithm’s updates are not only noisy, but can be severely biased.  Hence, [6] proposed a variant of TD that projects the iterated parameters onto a norm ball. This projection step imposes a uniform bound on the noise of gradient updates across time, which is needed for tractability. Later on, similar techniques have been extended to Q-leaning with linear function approximation [62, 63] and neural network function approximation [13, 22, 58]. Note that all these works **target the convergence of Q function values**, where the bounded gradient plays a key role.
>
> To remove the projection step under the Markovian observations, [47] is the first work that treats the TD-learning with linear function approximation as a linear stochastic approximation with Markovian noise and shows **the convergence of the parameters in terms of Lyapunov stability theory for linear ODEs by designing a suitable Lyapunov function.** Since the stochastic approximation-based technique tracks the drift of parameters rather than the Q function values, there is no need to track the gradient for parameter update. Instead, we may leverage other properties such as Lipshcitz continuity of the approximate linear functions, which we proved as in Lemmas 1-3 (lines 268-274). Later on, [15] extends the result to Q-learning with linear function approximation.
>
> Till now, the finite-time performance for Q-learning with neural network function approximation is unexplored, let alone our proposed Q-learning-based Whittle index with neural network function approximation (Neural-Q-Whittle). Our goal is to provide the first-ever finite-time convergence of Neural-Q-Whittle without the additional projection step by treating it as a two-timescale stochastic approximation under Markovian observations, which tracks the drift of parameters rather than Q function values.
>
> **Weakness #2:** ...whittle index definition...
>
> **Our Response:** We thank the reviewer for this insightful comment and sharp observation. We believe this is a misunderstanding here, mainly due to our unclear statements. We are sorry for this and will make it clear in the camera-ready version. Below, we provide some clarifications.
>
> In our paper, the Whittle index is still defined in the same manner as in conventional works [3, 23], which is the Lagrangian multiplier such that the Q values of action $0$ and $1$ are identical. With that being said, we have the following equation for action being $0$ and $1$,
> $$
> Q_n(s,0)+{\beta}=r_n(s,0)+\lambda+\sum_{s^\prime}p_n(s^\prime |s,0)V_n(s^\prime),
> $$
> $$
> Q_n(s,1)+{\beta}=r_n(s,1)+\sum_{s^\prime}p_n(s^\prime |s,1)V_n(s^\prime).
> $$
> Hence, we have
> $
> Q_n(s,0)-Q_n(s,1) =r_n(s,0)+\lambda+\sum_{s^\prime}p_n(s^\prime |s,0)V_n(s^\prime)-r_n(s,1)-\sum_{s^\prime}p_n(s^\prime |s,1)V_n(s^\prime), $
> which means that $\lambda^*\_n(s)$ is the value to make $Q\_n(s,0)-Q\_n(s,1)=0$.
> This leads to the definition of Whittle index in Equation (3), i.e.,
> $$
> \lambda\_n^*(s) = r\_n(s,1)+\sum_{s^\prime}p_n(s^\prime |s,1)V\_n(s^\prime)-r_n(s,0)-\sum_{s^\prime}p_n(s^\prime |s,0)V_n(s^\prime).
> $$
> This is exactly the same as Equation (4) in [23] and Equation (11) in [3]. Note that in current Eq. (3), the $Q_n(s,0)-Q_n(s,1)$ is redundant and should be removed. We thank the reviewer again for this comment and we will make this part clear in the camera-ready version.
>
> **Your Question #2:** ... step size...
>
> **Our Response:** Yes. As long as the stepsize sequences $\alpha$ and $\beta$ satisfy the conditions in line 119 and line 121, an almost-sure convergence for learning Whittle index by Q-learning has been characterized in [3]. However, in this paper, we aim to provide the first-ever finite-time convergence rate by carefully designing the stepsize sequences as in Theorem 1 (line 212), which achieves the best-known convergence speed $\mathcal{O}(1/k^{2/3})$ as the general linear and nonlinear 2TSA with i.i.d noise [19, 21].
>
> **Limitation #2:** ... 2-layer NN...
>
> **Our Response:** Thank you for this insightful comment. Unfortunately, we are afraid that we cannot agree with the reviewer's argument. The local linearization is a common trick to tackle the non-convexity of the original neural network function in finite-time analysis [13, 22, 58]. The major difference is that they characterize the convergence of Q function values while we track the drift of parameters. That is the fundamental reason why they need a projection step for parameter update while we do not require it. The local linearization technique can indeed be extended to multi-layer neural network function as in [58].
>
> The reason why we consider a two-layer NN is that the two-layer NN serves as the basis of multi-layer NN and it has strong representation power as the size $m$ goes large [13, 22].  The theoretical results of two-layer NN provide the fundamentals to characterize the result of multi-layer NN, just as what has been done in [13, 22].  In the current manuscript, our focus is to provide a fundamental theoretical analysis of a two-timescale  Q-learning-based Whittle index with neural network function approximation, which is already an open question and quite challenging, for which we make clear contributions as we summarize in the introduction. We thank the reviewer for pointing out this question, and extending the current framework with multiple layers is our future research.

---

> > ### Comment · Reviewer_ZyEQ · 2023-08-13
> > **Thank you for your response**
> >
> > I appreciate the authors for their detailed response, especially the response to my Question #2. It addresses my concern about the definition of Whittle index and I will update the rating. Thank you!

---

> > > ### Author Response · Authors · 2023-08-13
> > > **Thank you!**
> > >
> > > Thank you for your acknowledgement and raising the rating of our paper. Much appreciated!

---

### Official Review · Reviewer_rZoR · 2023-07-07

**Soundness:** 3 good
**Presentation:** 3 good
**Contribution:** 2 fair
**Rating:** 6
**Confidence:** 3

**Summary:**

This paper investigates the finite-time analysis of the Whittle index-based Q-learning policy for the RMAB problem under neural function approximation. The authors formulate the algorithm as a nonlinear two-time-scale stochastic approximation problem and present a convergence rate of $K^{2/3}$.

**Strengths:**

1. The Neural-Q-Whittle algorithm eliminates the projection step.
2. The paper provides finite-time analysis.
3. Simulation results are included to verify the convergence performance.

**Weaknesses:**

1. It is unclear whether the approximated Q-functions converge or not.
2. The errors diminish as $m\rightarrow \infty$, indicating the need for overparameterization.
3. The simulation setting is too simple to sufficiently demonstrate the advantages of neural approximation. Consideration of a larger state and action space is warranted.

**Questions:**

Since two-time-scale stochastic approximation has been extensively studied recently, could the authors clarify the novel main technical contributions of this paper?

**Limitations:**

More limitations of the approach should be added.

---

> ### Author Rebuttal · Authors · 2023-08-09
>
> Thank you very much for your review and constructive comments. Here we would like to address the reviewer's concerns and hope that can help raise the rating of our paper. The detailed responses are as follows:
>
> **Weakness #1:** It is unclear whether the approximated Q-functions converge or not.
>
> **Our Response:** Thank you for this question. Yes, the approximated Q-functions converge. Need to mention that due to the two-timescale nature for the updates of two coupled parameters (i.e., Q-function values and Whittle indices) in our proposed Neural-Q-Whittle, we focus on the convergence rate of these parameters rather than the convergence rate of approximated Q-functions as in [13, 22, 58]. Our key technique is to view Neural-Q-Whittle as a two-timescale stochastic approximation (2TSA) for finding the solution of suitable nonlinear equations. **Since we have theoretically shown that the parameters can converge to its unique optimal value in Theorem 2 when the neural network function $f$ is linearized as $f_0$,  the convergence of parameters in the original neural network function $f$ is established by adding the error between $f$ and $f_0$ introduced by the local linearaiztaion as in Theorem 1.  As long as $m\rightarrow\infty$, the learned parameters converge to the true optimal parameters with the convergence speed $\mathcal{O}(1/k^{2/3})$.**
>
> **Weakness #2:** ...$m\rightarrow \infty$ ...
>
> **Our Response:** Yes, that is true. The overparameterization is important in characterizing the convergence of Q-learning with neural network function approximation as in [13, 22, 58]. For a small $m$ value, there will always be a non-diminishing error in the convergence bound.
>
> **Weakness #3:** ...larger state and action space is warranted...
>
> **Our Response:** Thank you for your suggestion.  Since we are learning the Whittle index for restless multi-armed bandits (RMAB), the action space is two, i.e., 0 and 1 in the RMAB literature. In addition, the true Whittle index for the general RMAB problem is hard to be solved and may not have a closed-form solution. Hence, we consider a special queueing scenario of the RMAB problem where the state of each arm evolves as a controlled birth and death process as [M. Larrnaaga, U. Ayesta and I. M. Verloop, Dynamic Control of Birth-and-Death Restless Bandits: Application to Resource-Allocation Problems].  Per the reviewer's suggestion, we consider a larger state space, i.e., an arm with 51 states, which range from 0 to 50, representing the queue length. We randomly select a state and compare the learned Whittle index by our Neural-Q-Whittle with all other benchmarks in this paper.  For these experimental results, **Please kindly refer to the pdf in the General Response.** Again, it is clear that we make the same observations as we have in current experiments.  Since we will have an additional content page, we will add these results and discussions in the camera-ready version.  Finally, we note that although this work focuses on the theoretical convergence analysis of Q-learning based Whittle index under the neural network function approximation, it might be promising to implement it in real-world applications to fully leverage the strong representation ability of neural network functions, which serves as future investigation of this work.
>
> **Your Question #1:** Since two-time-scale stochastic approximation has been extensively studied recently, could the authors clarify the novel main technical contributions of this paper?
>
> **Our Response:** Thank you for this question and providing us an opportunity for clarifying. Two-timescale stochastic approximation (2TSA) has indeed been extensively studied as the references cited in our work [3, 18, 19, 20, 21, 38, 57] and many others. However, most of the current works focus on either linear or nonlinear with i.i.d noise [3, 18, 19, 21, 38, 57]. This may not be the real-world cases for reinforcement learning since the noise observed in each iteration is often not i.i.d. but Markovian. To our knowledge, the only work that considered Markovian noise is [20], which leveraged the same techniques in i.i.d noise setting as [19],  to explicitly characterize the covariance between the error caused by Markovian noise and the parameters’ residual error, leading to the convergence analysis much more intrinsic. Moreover, the analysis in [20] relies the strong monotone and smoothness of the considered functions. Unfortunately, these properties do not hold for the general neural network function as considered in our paper.  **Please kindly refer to the table in the pdf in General Response.**
>
> In this paper, we addressed two challenges that remain open in most of previous works. First, we removed the addition projection step introduced to stabilize the reinforcement learning due to Markovian noise. Second, we come up a new manner to deal with the non-linear neural network functions in a two- timescale stocahstic approximation framework, which is to deal with a local linearized neural network function first and then characterizing the error between the local linearization and the original neural network function.

---

> > ### Comment · Reviewer_rZoR · 2023-08-16
> >
> > Thanks for your response and clarifying some of my questions. I would like to increase the rating.

---

> > > ### Author Response · Authors · 2023-08-16
> > > **Thank you!**
> > >
> > > Thank you for your acknowledgement and raising the rating of our paper. Much appreciated!

---

### Official Review · Reviewer_KAmk · 2023-07-18

**Soundness:** 4 excellent
**Presentation:** 2 fair
**Contribution:** 3 good
**Rating:** 7
**Confidence:** 3

**Summary:**

This paper studies Whittle index-based Q-learning with neural network function approximations restless multi-armed bandits (RMAB) problem, which is a model-free low-complexity reinforcement learning (RL) heuristics for RMABs. Since state-action space of RMABs is exponentially growing with the number of arms, common Q-learning techniques as well as tabular Q-Whittle algorithm suffer from curse of dimensionality. Instead, the authors analyze a low-complexity neural network approximation method for Whittle index-based Q-learning under non-iid Markovian state-action observations, namely Neural-Q-Whittle. The authors formulate Neural-Q-Whittle as a nonlinear two-timescale stochastic approximation (2TSA) where the parameters of the neural network Q-function and the Whittle indices  are mutually coupled and the former is updated on a faster timescale than the latter. Leveraging a Lyapunov function method, the authors provide a finite-time convergence analysis of Neural-Q-Whittle with non-iid Markovian data. The analysis involves characterizing the error between two Lyapunov functions: one for the neural network Q-function and one for the linear approximation of the neural network.

**Strengths:**

This paper is original in the sense that it removes some of the limitations in the previous theoretical works on finite-time analysis of Whittle index, Q-learning, and nonlinear 2TSA and generalizes to neural network Q-function approximation in two-timescales under non-iid ergodic state-action process.  The theoretical analysis and proofs seem technically solid and mostly clear.


**Weaknesses:**

Since this work builds upon the limitations in the related past works, it would be easier to follow and appreciate the novelty and significance if you could provide a table comparing to related works in various aspects (eg. methods, assumptions, convergence rates, computational (time or space)  and sample complexities). It could also help with presentation if you give an informal version of your main theorem beforehand or simplify Theorem 1 using asymptotic notation and state the full version in the appendix.

I believe it should also be emphasized more that your convergence result relies on diminishing error bound of linear approximation of overparameterized neural networks.

Although this is mainly a theory paper, it would be interesting to see the experimental comparison to other methods such as Q-Whittle-LFA and references 13, 22, 58.



**Questions:**

1)I am not sure if I understand the sentence starting with "(Step 3)" between lines 204-207. Can you clarify what you mean exactly?

2) You might include the definition of "span" operator in equation 17 as a footnote.

3) Are h, g, y in Remark 4 supposed be h_0, g_0, y_0?

4) It is not exactly clear to me why it is desirable to update neural network parameters and the indices at two different time-scales. Do these parameters only converge under this case? Or do they converge faster compared to single timescale? Could you clarify the need for and the significance of two-timescale approximation?








**Limitations:**

The limitations are addressed within the text and mentioned as a future work in the conclusion (extension to multi-layer neural network).

---

> ### Author Rebuttal · Authors · 2023-08-09
>
> Thank you very much for your review and constructive comments, as well as giving the positive rating of our work. Here we would like to address the reviewer's concerns and hope that can help raise the rating of our paper. The detailed responses are as follows:
>
> **Weakness #1:** ...provide a table comparing to related works...simplify Theorem 1 using asymptotic notation ...
>
> **Our Response:** Thank you for this insightful comment and suggestion. Since this paper is the first to provide a finite-time convergence rate of the complex setting that learning the Whittle index by leveraging Q-learning with neural network function approximation, it is not straightforward and fair to compare the convergence rates, computational (time or space) and sample complexities with those references listed in our paper. However, to further distinguish the difference between our papers and per the reviewer's suggestion, we can still compare those works in terms of the considered setting, such as the noise model, whether an approximation was used or not, what is the timescale of the algorithm, and whether it aims at learning Whittle index. We summarize these in a table (**Please kindly refer to the pdf in the global response** ).  As for simplifying Theorem 1, we will end up with $\mathcal{O}(1/k^2)+\mathcal{O}(1/k^{2/3})+c$  with c being a constant value depending on $m$ and it goes to 0 as $m\rightarrow \infty$. However, we believe that presenting a full version in the main paper helps to better explain each term at the r.h.s. of Eq. (17). Since we will have an additional content page, we can include these discussions in the camera-ready version of this paper.
>
> **Weakness \#2:** ...emphasized more ...overparameterized ...
>
> **Our Response:** Thank you for this valuable suggestion. Indeed, the error bounds of linearization with the original neural network functions are controlled by the overparamterization value $m$, which has an impact on the global convergence as clearly shown in Theorem 1. We will emphasis this more in the camera-ready version.
>
> **Weakness \#3:** ... experimental comparison ...
>
> **Our Response:** Thank you for this suggestion. We indeed provided the comparison with Q-Whittle-LFA [57], WIQL [8] and QWIC [23] in supplementary material F (Figure 4 on page 26 in supplementary materials). The reason that [13, 22, 58] are not compared is that our proposed Neural-Q-Whittle in this paper aims to leverage Q-learning with neural network function approximation to learn Whittle index, which involves two coupled parameters, i.e., the Q function values and the Whittle indices. However, the algorithms in [13, 22, 58] are only Q-learning (or TD learning) with neural network function approximations, which only has one single parameter, i.e., the Q function values. To this end, we can only compare with those benchmarks which aim at learning Whittle indices.  Based on the reviewer's suggestion and since we will have an additional content page, we could add/move the experimental results in Figure 4 to the main paper in the camera-ready version.
>
> **Your Question #1:** ...Step 3...
>
> **Our Response:** Sorry for the confusion. We would like to explain this with the aid of Figure 1 on page 6 in the main paper. Our goal is to characterize the finite-time convergence of the two-timescale stochastic approximation (2TSA) defined in Eq. (9) where $h$ and $g$ are related with the true neural network function $f$. To do so, we define the Lyapunov function $M(\pmb{\theta}_k, \lambda_k)$ as in Eq. (15). However, it is challenging to directly finding the global optimum of the corresponding nonlinear equations due to the nonlinear neural network parameterization of Q-function in Neural-Q-Whittle. To tackle this challenge, we first provide a linearization $f\_0$ of the original neural network function $f$ and define a new Lyapunov function $\hat{M}(\pmb{\theta}\_k, \lambda\_k)$ as in Eq. (16). We then can study the convergence rate of the nonlinear 2TSA using this modified Lyapunov function with smoothness properties guaranteed due to linearization. Theorem 2 provides the convergence results for the surrogate linearized function  $f\_0$, which assumes that the 2TSA updates in Eq. (9) are based on the $f\_0$ function.  However, two coupled parameters $\pmb{\theta}$ and $\lambda$ in (9) are updated with respect to the true neural network function $f$, not $f\_0$. Hence, this further requires us to characterize the approximation errors between $f$ and $f_0$. When adding back this error to the result in Theorem 2, we finally get the desired result in Theorem 1.
>
> **Your Questions #2 and #3:** ..."span"... $h\_0, g\_0, y\_0$
>
> **Our Response:** Thank you for this suggestion and pointing out this typo. Yes, it is $h_0, g_0, y_0$, and we will fix them in the camera-ready version.
>
> **Your Question \#4:** ... two different time-scales...
>
> **Our Response:** Thank you for this insightful question. First, we would like to point out that it is not necessary to be two-timescale parameters.  However, in practice, for an update with two coupled variables with different sensitivities, it is usually empirically better to make one update quicker than the other, which has been widely studied as in [3,8, 19, 20, 21, 38] and reference therein.  As in our setting, the Whittle index depends on the value of Q functions, and thus it is natural to make Q functions update quicker than the Whittle index updates. This technique has also been used in [3, 57]. Second, from the perspective of theoretical performance analysis, this has been widely and theoretically studied in two-timescale stochastic approximation (2TSA) literature, e.g., [18, 19, 20, 21, 38],  with two learning rates, i.e., $\eta_{n,k}=o(\alpha_{n,k})$ (see lines 121 and 119). Indeed, by controlling these parameters as in two-timescale and as defined in Theorem 1 (line 212) , we achieve the best-known convergence speed $\mathcal{O}(1/k^{2/3})$.

---

> > ### Comment · Reviewer_KAmk · 2023-08-22
> >
> > Thank you for taking time and effort in addressing my questions and concerns and clarifying some of the confusing points. I am satisfied with your response and will keep my score.

---

> > > ### Author Response · Authors · 2023-08-22
> > > **Thank you!**
> > >
> > > Thank you for your acknowledgement and keeping the positive rating of our paper. Much appreciated!

---

### Official Review · Reviewer_bjcG · 2023-07-26

**Soundness:** 3 good
**Presentation:** 3 good
**Contribution:** 3 good
**Rating:** 6
**Confidence:** 3

**Summary:**

This paper provides a finite-time analysis of Neural-Q-Whittle. The authors propose Neural-Q-Whittle, a novel Whittle index-based Q-learning algorithm with neural network function approximation for RMAB. Their analysis leverages a Lyapunov drift approach to capture the evolution of two coupled parameters, and the nonlinearity in value function approximation further requires us to characterize the approximation error. They also conduct experiments to validate the convergence performance of Neural-Q-Whittle, and verify the sufficiency of their proposed condition for the stability of Neural-Q-Whittle.

**Strengths:**

1. This paper provides a non-asymptotic convergence rate analysis of Neural-Q-Whittle with two coupled parameters updated in two timescales under Markovian observations without the extra projection step.
2. The authors propose Neural-Q-Whittle, a novel Whittle index-based Q-learning algorithm with neural network function approximation for RMAB.
3. The authors establish the first finite-time analysis of Neural-Q-Whittle under Markovian observations. Their analysis leverages a Lyapunov drift approach to capture the evolution of two coupled parameters, and the nonlinearity in value function approximation further requires us to characterize the approximation error.
4. The writing is good.

**Weaknesses:**

Compared with existing works [13, 22, 58] for Q-learning with neural network function approximation, the technique contribution of this paper is kind of limited. It will be better if the authors could include additional parts to clarify the technical contribution of the paper, compared with prior works.

**Questions:**

Please refer to the Weaknesses section.

---

> ### Author Rebuttal · Authors · 2023-08-07
>
> Thank you very much for your review and constructive comments, as well as giving the positive rating of our work. Here we would like to address the reviewer's concerns and hope that can help raise the rating of our paper. The detailed responses are as follows:
>
> **Weakness #1:** Compared with existing works [13, 22, 58] for Q-learning with neural network function approximation, the technique contribution of this paper is kind of limited. It will be better if the authors could include additional parts to clarify the technical contribution of the paper, compared with prior works.
>
> **Our Response:** Thank you for this insightful comment and providing us a chance to further clarify the contributions of this paper.
>
> Compared with existing works [13, 22, 58] for Q-learning with neural network function approximation, the first fundamental difference of our proposed Neural-Q-Whittle lies in the algorithm framework. Existing works [13, 22, 58] aimed to learn Q-function values with neural network function approximation, while our proposed Neural-Q-Whittle aims to learn the Whittle index for restless multi-armed bandits (RMAB) problem by leveraging Q-learning with neural network function approximation. In particular, our Neural-Q-Whittle involves a two-timescale update between two coupled parameters, i.e., Q-function values $Q(s,a), \forall s,a$ and Whittle indices $\lambda(s), \forall s$ as defined in Eq. (4)-(5). **This renders existing finite-time analysis in [13, 22, 58] not applicable to our Neural-Q-Whittle** due to the fact that [13, 22, 58] only contained a single-timescale update only on Q-function values $Q(s,a), \forall s, a.$ **Hence, it requires a fundamental different technique (i.e., two-timescale analysis) to establish the finite-time convergence of our proposed Neural-Q-Whittle with two-coupled parameters.**
>
> Second, due to the Markovian observations of Q-learning at each iteration, conventional single-timescale update Q-learning with neural network function approximation [13, 22, 58] **required an additional projection step for the update of parameters of neural network function** so as to guarantee the boundedness between the unknown parameter at any time step with the initialization, which stabilizes the updates. However, this in some cases is impractical. **Hence, how to remove this additional projection step is an open research problem in the community.** One possible way to remove the projection step is to treat the Q-learning (or TD learning) update as one-timescale stochastic approximation (SA) as in [15, 47]. However, these results only apply to Q-learning with linear function approximation. To the best of our knowledge, the results for single-timescale Q-learning with neural network function approximation is even unknown, let alone the two-timescale Neural-Q-Whittle learning that is considered in this paper. Therefore,  a natural question that arises is: **Is it possible to provide a non-asymptotic convergence rate analysis of Neural-Q-Whittle with two coupled parameters updated in two timescales under Markovian observations without the extra projection step?** Indeed, we provide an affirmative answer to this question in this paper.
>
> With that being said, our major contribution in this paper is to establish the first-ever finite-time analysis of Neural-Q-Whittle under Markovian observations. **Due to the two-timescale nature for the updates of two coupled parameters (i.e., Q-function values and Whittle indices) in Neural-Q-Whittle, we focus on the convergence rate of these parameters rather than the convergence rate of approximated Q-functions as in [13, 22, 58].** Our key technique is to view Neural-Q-Whittle as a two-timescale stochastic approximation (2TSA) for finding the solution of suitable nonlinear equations. Different from recent works on finite-time analysis of a general 2TSA [20] or with linear function approximation [57], the nonlinear parameterization of Q-function in Neural-Q-Whittle under Markovian observations imposes significant difficulty in finding the global optimum of the corresponding nonlinear equations. To mitigate this, we first approximate the original neural network function with a collection of local linearization and focus on finding a surrogate Q-function in the neural network function class that well approximates the optimum. Our finite-time analysis then requires us to consider two Lyapunov functions that carefully characterize the coupling between iterates of Q-function values and Whittle indices, with one Lyapunov function defined with respect to the true neural network function, and the other defined with respect to the locally linearized neural network function. We then characterize the errors between these two Lyapunov functions. Putting them together, we prove that Neural-Q-Whittle achieves a convergence in expectation at a rate
> $\mathcal{O}(1/k^{2/3})$, where $k$ is the number of iterations.

---

> > ### Comment · Reviewer_bjcG · 2023-08-17
> >
> > Thank you very much for your response and clarification.

---

> > > ### Author Response · Authors · 2023-08-17
> > > **Thank you!**
> > >
> > > Thank you for your acknowledgement and keeping the positive rating of our paper. Much appreciated!

---

### Author Rebuttal · Authors · 2023-08-09

The attached Pdf contains a table for **Reviewer KAmk** and **Reviewer rZoR**, and a figure for **Reviewer rZoR.** The detailed response to the corresponding comments are provided below in our rebuttal to each reviewer.

---

### Decision · Program_Chairs · 2023-09-21

**Decision:**

Accept (poster)

**Comment:**

Reviewers unanimously agree on acceptance.
Please consider the authors comments regarding several confusing parts in the manuscript, and be sure to fix them for the camera-ready version.